# Public knowledge, practices, and awareness of antibiotics and antibiotic resistance in Myanmar: The first national mobile phone panel survey

**Shinsuke Miyano**[1,2]*, **Thi Thi Htoon**[3], **Ikuma Nozaki**[2], **Eh Htoo Pe**[3], **Htay Htay Tin**[4]

**1** Advisor for Infectious Disease Control and Laboratory Services, Japan International Cooperation Agency (JICA), Yangon, Myanmar, **2** Bureau of International Health Cooperation, National Center for Global Health and Medicine (NCGM), Tokyo, Japan, **3** Department of Medical Services, National Health Laboratory (NHL) / National AMR Coordinating Center (NCC), Ministry of Health, Yangon, Myanmar, **4** University of Medical Technology, Yangon, Ministry of Health, Yangon, Myanmar

* s-miyano@it.ncgm.go.jp

**Data Availability Statement:** Data cannot be shared publicly because these data are belonging to the National Programme in Myanmar. The data will be available following publication and will be

## Abstract

In 2017, the Myanmar National Action Plan for Containment of Antimicrobial Resistance (AMR) (2017–2022) was endorsed by the Ministry of Health and Sports, Myanmar; one of its objectives was to increase public awareness of AMR to accelerate appropriate antibiotic use. This survey aimed to assess the public knowledge, practices and awareness concerning antibiotics and AMR awareness among adults in Myanmar. We conducted a nationwide cross-sectional mobile phone panel survey in January and February 2020. Participants were randomly selected from the mobile phone panel in each of three groups stratified by gender, age group, and residential area urbanity; they were interviewed using a structured questionnaire. Collected data were weighted based on the population of each stratum from the latest national census and analyzed using descriptive and inferential statistics. Two thousand and forty-five adults from 12 regions and states participated in this survey. Overall, 89.5% of participants had heard about antibiotics; however, only 0.9% provided correct answers to all five questions about antibiotics, whereas 9.7% provided all incorrect answers. More than half of participants (58.5%) purchased antibiotics without a prescription, mainly from medical stores or pharmacies (87.9%); this was more frequent in age group (18–29 years) and those in rural areas (p = 0.004 and p < 0.001, respectively). Only 56.3% were aware of antibiotic resistance and received their information from medical professionals (46.3%), family members or friends (38.9%), or the media (26.1%). Less than half (42.4%) knew that antibiotics were used in farm animals. Most did not know that using antibiotics in farm animals could develop resistance (73.2%) and is banned for the purposes of growth stimulation (64.1%). This survey identifies considerable gaps in the knowledge, practices, and awareness about antibiotics among the general population in Myanmar. Continuous public education and awareness campaigns must be urgently conducted to fulfill these gaps, which would aid in promoting antibiotic stewardship, leading to combating AMR in Myanmar.

shared with those who gets approval from the Department of Medical Research, Ministry of Health, Myanmar for researchers who meet the criteria for access to confidential data. For data access, please contact the Ethics Review Committee (ercdmr2015@gmail.com).

**Funding:** The corresponding author and the first author received grants from the National Center for Global Health and Medicine, Japan (19A09 and 20A07) (https://www.ncgm.go.jp/100/010/index.html), and the Projects for the Advisor for Infectious Disease Control and Laboratory Services, Japan International Cooperation Agency, Myanmar (FYI2019 and FYI2020) (https://www.jica.go.jp/myanmar/english/index.html).

**Competing interests:** The authors have declared that no competing interests exist.

## Introduction

Antimicrobial resistance (AMR) is a major threat to the clinical efficacy of antimicrobial medicines; as such, it is a critical challenge to global public health. In 2019, there were an estimated 4.95 million deaths associated with bacterial AMR [1]. A global report estimated that without further action, by 2050, 10 million people per year will be dead and 100 trillion USD worth of global production will be lost due to AMR [2]. In 2015, the World Health Assembly endorsed the Global Action Plan on AMR and urged all Member States to develop relevant national action plans within two years [3].

Antibiotics are the most familiar and commonly used antimicrobials among the general population. The accumulation of resistance to antibiotics is an inescapable consequence of the administration of antibiotics [4], and the misuse and abuse of antibiotics have drastically increased the risk of developing resistance [5, 6]. Even though new antibiotics have been developed, if the speed of the development of resistance is faster than that of the development of new antibiotics, then there will be few antibiotic choices for treatment of bacterial infection. Thus, to minimize the risk of AMR, it is critical that knowledge about antibiotics, appropriate antibiotic use, and awareness of AMR is increased among the general public. In 2015, the World Health Organization (WHO) conducted a multi-country public awareness survey on antibiotic resistance in 12 countries, including two member states from each WHO region [7]. Other countries have also conducted similar surveys to understand their populations' knowledge and awareness, such that they can plan and implement evidence-based interventions in each setting [8–11].

In the WHO South-East Asian region, only three countries have conducted nationwide surveys to assess public awareness of antibiotic resistance; India and Indonesia implemented the WHO survey by online [7], whereas Thailand conducted a survey via the creation of a module to measure awareness as part of the 2017 national health welfare survey [8]. These surveys highlighted insufficient levels of knowledge around the appropriate use of antibiotics and awareness of the issue of antibiotic resistance among the people.

In Myanmar, according to the latest national AMR surveillance data in 2017, high levels of antibiotic resistance bacteria were reported; methicillin-resistant *Staphylococcus aureus* (40%), extended spectrum beta-lactamase producing *Enterobacteriaceae* (48%), carbapenem-resistant *Acinetobacter* species (9%) and *Pseudomonas* species (19%), and vancomycin-resistant *Enterococcus* species (15%) [12]. To combat against AMR, in 2017, the Ministry of Health and Sports, Myanmar, published the National Action Plan for Containment of AMR in Myanmar 2017–2022 [13]. The plan consists of five strategic objectives, in line with those of the global action plan on AMR. One of these strategic objectives is to improve public awareness and understanding of AMR via effective communication, education, and training. Within this objective, one of the planned actions is the creation of an evidence-based public communications program, which is set to be implemented on a national scale to improve awareness of AMR among the public and professionals; nationwide evidence-based awareness campaigns are set to be enacted with regular monitoring and evaluation by 2022. Nonetheless, there is currently no data directly evaluating the public awareness and perception of AMR in Myanmar.

Many countries gather population-based survey data through traditional face-to-face household surveys [14]. Although traditional household surveys can include all individuals living in accessible locations, they require more time and resources than mobile phone panel surveys (MPPSs) [15]. In recent years, especially during the COVID-19 pandemic, researchers have been increasingly interested in using modern technologies to gather high-quality and high-frequency survey data related to population perceptions [16, 17]. With the availability of inexpensive phone handsets and the rapidly growing network coverage in many countries,

mobile phones have attracted considerable attention as a new tool for collecting high-frequency survey data using fewer resources and lower costs [18]. In Myanmar, the possession rate of mobile phones has dramatically increased. For instance, in 2016, a national baseline survey of the Asian region revealed that 61% of the Myanmar population aged 15–65 years own a mobile phone, and 78% of these individuals own smartphones; additionally, 41% of all non–mobile phone holders had plans to get connected in the future [19].

Thus, this study aimed to assess public knowledge and practices concerning antibiotics, as well as awareness of antibiotic resistance, to provide baseline evidence, using the MPPS method, and identify gaps related to the deployment of strategic awareness-raising activities in Myanmar.

## Materials and methods

### Survey design and population

This study was a nationwide cross-sectional survey that included individuals aged 18 years and older. A mobile phone panel survey (MPPS) methodology was adopted to collect representative data nationwide, using mobile phone interviews.

The mobile phone panel we used was regularly updated by a survey office that cooperated with government surveys and was composed of 200,000 contacts. Field workers across the country interviewed potential candidates to the panel in person at their homes, collected their basic demographic information and received their permission to be included in the panel to respond to several MPPSs on various topics. The global guidelines for MPPSs recommended a sample size of 500 individuals for national surveys [15]; however, since this survey required calls for disaggregation by three groups stratified by gender (male and female), age group (18–29, 30–39, 40–49, and $\geqq$50 years old), and residential area (urban and rural), sample size calculations had to be performed separately for each group, and a required sample size of 2,000 individuals was deemed to be sufficient. Since the survey used mobile phone panel data, the final sample size required accounting for attrition during the planned survey series. The national AMR program planned to implement two survey rounds to monitor the future situation, with an assumed attrition of 5.0% in each round. Assuming that attrition is random, the eventual sample size was estimated to be 2,216 (calculated as $2,000/(1–0.05)^2$) [15].

### Data collection

The survey was conducted from January 17 to February 14, 2020. Data were collected using a structured interview methodology called interactive voice response. At the call center, nineteen trained interviewers called participants that had been randomly selected from the panel, collected basic information on general characteristics such as age, birthday, gender, and residential area, and posed questions using a fixed questionnaire. When the participant dropped out of the interview due to poor phone connection and/or had a problem in answering the questions, the interviewer replaced with a different participant with similar demographics in the panel. For quality data collection, the quality control team randomly selects 30% of interview results per interviewer and investigates by listening to the recordings or calling the participant for verification. We translated the questionnaire that is commonly used in English by the WHO [7] and the European Union Commission [20] into the local language, using back translation method; we later modified some questions after pretesting the translated survey with 40 local individuals. The questionnaire was composed of four sections: i) knowledge of antibiotics; ii) usage of antibiotics; iii) knowledge of antibiotic resistance; and iv) awareness of antibiotic use in farm animals (S1 File). We confirmed the participants' recognition of "antibiotics"

in the first question and then asked all the other questions only to those who knew antibiotics in this survey.

## Data analysis

All data were double-entered into a Microsoft Excel 2017 spreadsheet. Data were analyzed statistically using Statistical Package for the Social Sciences (SPSS) version 24. When calculating the percentage of participants who provided a particular response to each question, the sampling design and sampling weight of each participant were considered to ensure representativeness and unbiased results. In terms of the sample weight, the weighting factor was calculated using population data for each gender, age group, and residential area urbanity based on the latest National Census, conducted in 2014. The Clopper–Pearson exact method was used to calculate 95% confidence intervals (CIs) for the estimated percentage of each group. Pearson's Chi-square test was used to examine differences between groups. A p-value $< 0.05$ was considered statistically significant.

## Ethical considerations

The interviewers verbally explained the detailed survey objectives and procedures to the selected participants on the phone. Verbal informed consent was obtained from all selected participants and recorded; however, the participants' names and any identifying information were not recorded. The research protocol was approved by the National Center for Global Health and Medicine, Japan (NCGM-G-003240-00) and the Institutional Review Board for Biomedical Research, Department of Medical Research, Ministry of Health and Sports, Myanmar (DMR/2019/143).

## Results

Two thousand and forty-five people participated in the survey, in which the average duration of each interview was 7.53 minutes. The non-response rate, including refusal, poor mobile phone signal, or wrong phone number, was 64.0% (9,856/15,399 contacts). After considering the sampling design and sampling weight of each participant, 52.5% of participants were female, the mean age was 39.4 years (range 18–80), 33.3% of participants were under 30 years old, and 22.7% of participants were 50 years old or older. Participants were originated from 12 of 15 regions and states, and 68.5% of participants lived in a rural area (Table 1).

### Recognition of antibiotics

Approximately half (51.0%) of the participants knew antibiotics; however, given that there is no exact word for "antibiotics" in the Myanmar language, the interviewers cited examples of the names of antibiotics that are commonly used in Myanmar. Consequently, 89.5% of participants recognized antibiotics. Although individuals in rural areas (p < 0.001) and those 29 years of age or younger (p = 0.02) had a lower recognition of antibiotics, there was no statistical difference between males and females (Table 2).

### Knowledge of antibiotics

Among the participants who recognized antibiotics as a medicine, most believed that antibiotics could kill viruses (72.6%) and are effective against colds and flu (73.5%); these beliefs were significantly greater among individuals in urban areas and those 18–29 years of ages (Table 3). More than half of participants knew that the unnecessary or inappropriate use of antibiotics could result in ineffective treatment or resistance (60.6%); this proportion was significantly

**Table 1. Participants' demographic characteristics in Myanmar, 2020 (N = 2,045).**

| Characteristics | n (weighted) | % (estimated) |
|---|---|---|
| **Gender** | | |
| female | 1,074 | 52.5 |
| male | 971 | 47.5 |
| **Age group** | | |
| 18–29 | 680 | 33.3 |
| 30–39 | 491 | 24.0 |
| 40–49 | 410 | 20.0 |
| 50- | 464 | 22.7 |
| **Urbanity** | | |
| urban | 645 | 31.5 |
| rural | 1,400 | 68.5 |
| **Region and State** | | |
| Ayeyarwady | 222 | 10.9 |
| Bago | 193 | 9.4 |
| Kachin | 67 | 3.3 |
| Kayin | 55 | 2.7 |
| Magway | 142 | 6.9 |
| Mandalay | 279 | 13.6 |
| Mon | 88 | 4.3 |
| Nay Pyi Taw | 55 | 2.7 |
| Sagaing | 182 | 8.9 |
| Shan | 218 | 10.7 |
| Tanintharyi | 61 | 3.0 |
| Yangon | 463 | 22.6 |

higher in urban areas (p < 0.001). Less than half of participants (39.1%) knew that taking antibiotics often has side effects, such as diarrhea; females (p = 0.02) and older individuals (p < 0.001) more frequently provided correct answers to this question. Approximately three-quarters of participants knew that antibiotics are not equal to anti-inflammatory drugs such as

**Table 2. Participants' recognition of antibiotics (N = 2,045).**

| Characteristics | N (weighted) | Recognition of antibiotics | | | |
|---|---|---|---|---|---|
| | | n (weighted) | % † (estimated) | 95% CI | p-value |
| **Overall** | 2,045 | 1,830 | 89.5 | 88.1–90.7 | - |
| **Gender** | | | | | |
| female | 1,074 | 974 | 90.7 | 88.8–92.3 | 0.07 |
| male | 971 | 856 | 88.1 | 86.0–90.0 | |
| **Age group** | | | | | |
| 18–29 | 680 | 588 | 86.5 | 83.7–88.8 | 0.02 * |
| 30–39 | 491 | 444 | 90.4 | 87.5–92.7 | |
| 40–49 | 410 | 377 | 92.0 | 88.9–94.2 | |
| 50- | 464 | 421 | 90.8 | 87.7–93.1 | |
| **Urbanity** | | | | | |
| urban | 645 | 603 | 93.5 | 91.3–95.2 | <0.001 * |
| rural | 1,400 | 1,227 | 87.7 | 85.8–89.3 | |

*refer to statistically significance

†row percentage

**Table 3. Five true/false questions used to assess the survey participants' knowledge of antibiotics (N = 1,830).**

| Characteristics | Antibiotics can kill viruses (False) weighted n (%) † | | | | Antibiotics are effective against cold and flu (False) weighted n (%) † | | | | Unnecessary or inappropriate use of antibiotics can result in ineffective treatment or resistance (True) weighted n (%) † | | | |
|---|---|---|---|---|---|---|---|---|---|---|---|---|
| | Correct | Incorrect | Don't know | p-value | Correct | Incorrect | Don't know | p-value | Correct | Incorrect | Don't know | p-value |
| **Overall** | 213 (11.7) | 1,329 (72.6) | 287 (15.7) | - | 279 (15.3) | 1,345 (73.5) | 205 (11.2) | - | 1,110 (60.6) | 542 (29.6) | 178 (9.7) | - |
| **Gender** | | | | | | | | | | | | |
| female | 99 (10.2) | 725 (74.5) | 150 (15.4) | 0.08 | 142 (14.6) | 726 (74.5) | 106 (10.9) | 0.58 | 598 (61.4) | 286 (29.3) | 90 (9.3) | 0.69 |
| male | 114 (13.4) | 604 (70.6) | 138 (16.1) | | 137 (16.0) | 619 (72.4) | 99 (11.6) | | 512 (59.8) | 256 (30.0) | 88 (10.3) | |
| **Age group** | | | | | | | | | | | | |
| 18–29 | 85 (14.4) | 427 (72.6) | 76 (13.0) | 0.001 * | 136 (23.1) | 386 (65.7) | 66 (11.2) | <0.001 * | 362 (61.6) | 172 (29.3) | 53 (9.1) | 0.91 |
| 30–39 | 56 (12.5) | 327 (73.7) | 61 (13.7) | | 70 (15.7) | 328 (74.0) | 46 (10.3) | | 263 (59.2) | 140 (31.5) | 41 (9.3) | |
| 40–49 | 43 (11.3) | 273 (72.3) | 62 (16.5) | | 31 (8.3) | 297 (78.9) | 49 (12.9) | | 226 (59.9) | 110 (29.2) | 41 (10.9) | |
| 50- | 30 (7.2) | 303 (71.9) | 88 (20.8) | | 43 (10.1) | 334 (79.2) | 45 (10.7) | | 259 (61.5) | 120 (28.4) | 43 (10.1) | |
| **Urbanity** | | | | | | | | | | | | |
| urban | 65 (10.7) | 482 (80.0) | 56 (9.3) | <0.001 * | 93 (15.4) | 470 (77.9) | 40 (6.7) | <0.001 * | 395 (65.6) | 175 (29.0) | 33 (5.4) | <0.001 * |
| rural | 149 (12.1) | 847 (69.0) | 231 (18.9) | | 186 (15.2) | 876 (71.4) | 165 (13.5) | | 714 (58.2) | 367 (29.9) | 145 (11.9) | |

| Characteristics | Taking antibiotics has side-effects such as diarrhhea (True) weighted n (%) † | | | | Antibiotics are not equal to anti-inflammatory drugs such as painkillers or antipyretics (True) weighted n (%) † | | | | | | | |
|---|---|---|---|---|---|---|---|---|---|---|---|---|
| | Correct | Incorrect | Don't know | p-value | Correct | Incorrect | Don't know | p-value | | | | |
| **Overall** | 716 (39.1) | 764 (41.8) | 350 (19.1) | - | 1310 (71.6) | 303 (16.6) | 217 (11.9) | - | | | | |
| **Gender** | | | | | | | | | | | | |
| female | 409 (42.0) | 382 (39.3) | 182 (18.7) | 0.02 * | 690 (70.8) | 163 (16.7) | 122 (12.5) | 0.62 | | | | |
| male | 307 (35.8) | 382 (44.6) | 168 (19.6) | | 620 (72.4) | 141 (16.4) | 95 (11.1) | | | | | |
| **Age group** | | | | | | | | | | | | |
| 18–29 | 200 (33.9) | 278 (47.3) | 110 (18.7) | <0.001 * | 429 (73.0) | 103 (17.5) | 56 (9.5) | 0.24 | | | | |
| 30–39 | 162 (36.5) | 185 (41.8) | 96 (21.7) | | 313 (70.5) | 68 (15.3) | 63 (14.2) | | | | | |
| 40–49 | 159 (42.1) | 153 (40.5) | 66 (17.4) | | 263 (69.6) | 61 (16.2) | 53 (14.1) | | | | | |
| 50- | 196 (46.5) | 148 (35.2) | 77 (18.4) | | 305 (72.4) | 71 (16.9) | 45 (10.7) | | | | | |
| **Urbanity** | | | | | | | | | | | | |
| urban | 219 (36.2) | 275 (45.6) | 110 (18.2) | 0.07 | 461 (76.5) | 91 (15.1) | 51 (8.4) | 0.002 * | | | | |
| rural | 498 (40.5) | 489 (39.9) | 240 (19.6) | | 849 (69.2) | 212 (17.3) | 166 (13.5) | | | | | |

*refer to statistically significance

†row percentage

painkillers or antipyretics (71.6%); knowledge of this was greater in individuals from urban areas (p = 0.002). Overall, for all five true/false questions, 0.9% of participants provided only correct answers, whereas 9.7% provided only incorrect answers. Approximately one-third of participants provided three or more correct answers (32.7%).

## Antibiotic use practices

We noted that 54.7% of participants who recognized antibiotics as medicine had previously used antibiotics. Females (p = 0.04), individuals from urban areas (p = 0.001), and those aged 50 years and older (p < 0.001) had significantly more experience taking antibiotics, whereas those aged 18–29 years had less experience (Table 4). Approximately one-third of participants (36.8%) had used antibiotics in the previous six months. More than half of participants (58.5%) purchased antibiotics without a prescription; the frequency of this practice was significantly higher in individuals aged 18–29 years (p < 0.001) and those living in rural areas (p = 0.004). Most participants who had used antibiotics obtained them from a medical store or pharmacy (87.9%), stall or hawker (4.0%), or family members or friends (3.0%). Although only 17.7% of participants stopped taking their antibiotics after completion of the full course, more than half of all participants (62.5%) stopped taking antibiotics when they felt better. Individuals who obtained antibiotics with a prescription was more likely to complete the full course of treatment (p < 0.001).

## Awareness of antimicrobial resistance (AMR)

The concept of AMR was recognized by more than half of all participants (56.3%), with proportions being significantly higher in those from urban areas, compared to rural areas (63.4%

**Table 4. Antibiotic use practices (N = 1,830).**

| Characteristics | N (weighted) | Past experience taking antibiotics | | | | Purchased antibiotics without a prescription | | | | Stop taking antibiotics when feeling better | | | |
|---|---|---|---|---|---|---|---|---|---|---|---|---|---|
| | | n (weighted) | % † (estimated) | 95% CI | p-value | n (weighted) | % † (estimated) | 95% CI | p-value | n (weighted) | % † (estimated) | 95% CI | p-value |
| **Overall** | 1,830 | 1,001 | 54.7 | 52.4–57.0 | - | 586 | 58.5 | 55.5–61.6 | - | 625 | 62.5 | 59.4–65.4 | - |
| **Gender** | | | | | | | | | | | | | |
| female | 974 | 555 | 57.0 | 53.9–60.1 | 0.04 * | 317 | 57.1 | 53.0–61.2 | 0.27 | 360 | 64.8 | 60.8–68.7 | 0.1 |
| male | 856 | 445 | 52.0 | 48.6–55.3 | | 269 | 60.4 | 55.8–64.9 | | 265 | 59.6 | 54.9–64.0 | |
| **Age group** | | | | | | | | | | | | | |
| 18–29 | 588 | 265 | 45.0 | 41.1–49.1 | <0.001 * | 172 | 65.0 | 59.0–70.4 | 0.004 * | 175 | 65.5 | 60.1–71.5 | 0.62 |
| 30–39 | 444 | 257 | 57.8 | 53.2–62.4 | | 160 | 62.2 | 56.2–68.0 | | 155 | 60.3 | 54.2–66.1 | |
| 40–49 | 377 | 212 | 56.2 | 51.2–61.2 | | 119 | 56.2 | 49.4–62.6 | | 129 | 60.8 | 54.1–67.2 | |
| 50- | 421 | 268 | 63.6 | 59.0–68.1 | | 134 | 50.2 | 44.1–55.9 | | 166 | 62.1 | 56.0–67.5 | |
| **Urbanity** | | | | | | | | | | | | | |
| urban | 603 | 362 | 60.1 | 56.1–63.9 | 0.001 * | 177 | 49.0 | 43.8–54.0 | <0.001 * | 230 | 63.7 | 58.5–68.3 | 0.55 |
| rural | 1,227 | 638 | 52.0 | 49.2–54.8 | | 408 | 63.9 | 60.1–67.6 | | 395 | 61.9 | 58.1–65.6 | |

*refer to statistically significance

†row percentage

**Table 5.** Awareness of antimicrobial resistance (AMR) (N = 1,830).

| Characteristics | N (weighted) | Awareness of AMR weighted n (%) | | | | Information on AMR changed your practice weighted n (%) | | | |
|---|---|---|---|---|---|---|---|---|---|
| | | n (weighted) | % † (estimated) | 95% CI | p-value | n (weighted) | % † (estimated) | 95% CI | p-value |
| **Overall** | 1,830 | 1,029 | 56.3 | 53.9–58.5 | - | 674 | 65.5 | 62.5–68.3 | - |
| **Gender** | | | | | | | | | |
| female | 974 | 563 | 57.8 | 54.7–60.9 | 0.15 | 378 | 67.1 | 63.2–70.9 | 0.22 |
| male | 856 | 466 | 54.5 | 51.1–57.7 | | 296 | 63.6 | 59.1–67.8 | |
| **Age group** | | | | | | | | | |
| 18–29 | 588 | 280 | 47.7 | 43.6–51.7 | <0.001 * | 192 | 68.3 | 62.9–73.7 | 0.41 |
| 30–39 | 444 | 266 | 60.0 | 55.3–64.4 | | 179 | 67.3 | 61.4–72.7 | |
| 40–49 | 377 | 213 | 56.6 | 51.5–61.4 | | 135 | 63.6 | 56.7–69.6 | |
| 50- | 421 | 270 | 64.0 | 59.4–68.6 | | 168 | 62.4 | 56.3–67.8 | |
| **Urbanity** | | | | | | | | | |
| urban | 603 | 382 | 63.4 | 59.4–67.1 | <0.001 * | 232 | 60.8 | 55.8–65.5 | 0.02 * |
| rural | 1227 | 647 | 52.7 | 49.9–55.5 | | 442 | 68.3 | 64.6–71.8 | |

*refer to statistically significance

†column percentage

vs. 52.7%, p < 0.001) and individuals aged 50 years and older, compared to 18–29 year olds (64.0% vs. 47.7%, p < 0.001; Table 5). Sources of information on AMR were doctors or nurses (46.3%), family member or friends (38.9%), media (26.1%), pharmacists (5.0%), and other sources (7.0%). In terms of media, TV (35.8%) and Facebook (21.4%) were the main sources of information on AMR. Most participants (65.5%) confirmed that the information they received about AMR had improved their practices related to the use of antibiotics; the proportion of individuals who agreed with this statement was significantly higher among those in rural areas, compared to urban areas (68.3% vs. 60.8%, p = 0.02).

## Awareness of antibiotic use and resistance in farm animals

Less than half of all participants (42.4%) knew that antibiotics were used to treat sick animals on farms (Table 6). Furthermore, only approximately one-third of participants knew that providing antibiotics to farm animals could develop resistance and is banned for the mere purpose of growth stimulation in Myanmar (26.8% and 35.9%, respectively). Nonetheless, there were no statistical differences across groups, other than those older individuals demonstrated a better understanding of the regulation on antibiotic use as a growth stimulator in farm animals (p = 0.001).

## Discussion

The current study was the first national survey to measure public knowledge and awareness of antibiotics and antibiotic resistance in Myanmar. Only a few similar studies had been conducted at the national level in low- and middle-income countries, including the neighboring countries of Thailand, Vietnam, India, and Indonesia [7, 8], whereas most similar studies had been conducted in high-income countries [10, 21–27]. Using a mobile phone panel, the large number of recruited participants allowed for accessing a variety of the people of different gender, age group, and residential areas, and improved sample representativeness in the survey of the Myanmar public, including urban and rural areas. The survey revealed gaps in antibiotics knowledge and indicated insufficient awareness of antibiotic resistance among the general public, seemingly leading to inappropriate antibiotic use practices.

**Table 6. Awareness of antibiotic use and resistance in farm animals (N = 1,830).**

| Characteristics | N (weighted) | Awareness of antibiotic use in farm animals weighted n (%) † | | | | Awareness of antibiotic resistance in farm animals weighted n (%) † | | | | Antibiotic use is banned for the mere purpose of growth stimulation in farm animals weighted n (%) † | | | |
|---|---|---|---|---|---|---|---|---|---|---|---|---|---|
| | | n (weighted) | % (estimated) | 95% CI | p-value | n (weighted) | % (estimated) | 95% CI | p-value | n (weighted) | % (estimated) | 95% CI | p-value |
| **Overall** | 1,830 | 777 | 42.4 | 40.2–44.7 | - | 490 | 26.8 | 24.8–28.9 | - | 656 | 35.9 | 33.7–38.1 | - |
| **Gender** | | | | | | | | | | | | | |
| female | 974 | 400 | 41.1 | 38.0–44.2 | 0.22 | 244 | 25.0 | 22.4–27.9 | 0.08 | 359 | 36.9 | 33.9–39.9 | 0.34 |
| male | 856 | 376 | 43.9 | 40.6–47.3 | | 246 | 28.8 | 25.8–31.9 | | 297 | 34.7 | 31.6–37.9 | |
| **Age group** | | | | | | | | | | | | | |
| 18–29 | 588 | 257 | 43.7 | 39.8–47.7 | 0.44 | 158 | 26.8 | 23.4–30.6 | 0.88 | 175 | 29.8 | 26.2–33.6 | 0.001 * |
| 30–39 | 444 | 196 | 44.3 | 39.6–48.8 | | 116 | 26.0 | 22.3–30.4 | | 162 | 36.5 | 32.1–41.1 | |
| 40–49 | 377 | 158 | 41.9 | 37.0–46.9 | | 98 | 25.9 | 21.8–30.7 | | 144 | 38.1 | 33.4–43.2 | |
| 50- | 421 | 166 | 39.3 | 34.9–44.2 | | 119 | 28.3 | 24.2–32.8 | | 175 | 41.6 | 37.0–46.3 | |
| **Urbanity** | | | | | | | | | | | | | |
| urban | 603 | 236 | 39.1 | 35.3–43.1 | 0.06 | 156 | 25.8 | 22.5–29.5 | 0.54 | 231 | 38.3 | 34.5–42.3 | 0.12 |
| rural | 1227 | 541 | 44.1 | 41.3–46.9 | | 334 | 27.2 | 24.8–29.8 | | 425 | 34.7 | 32.0–37.3 | |

*refer to statistically significance

†row percentage

The survey identified a significant gap in antibiotics knowledge among individuals in Myanmar. More than two-thirds of participants provided two or more incorrect answers to the five true/false questions used to assess their antibiotics knowledge; this implies that the people do not correctly understand antibiotics, although most recognize antibiotics as a medicine to treat infectious diseases. While the majority of participants understood that antibiotics could not be used as painkillers nor antipyretics, and that inappropriate antibiotic use decreases their effectiveness, compared to individuals in neighboring countries like Vietnam (36%), Indonesia (33%), Thailand (20%) and India (20%), people from Myanmar were less likely to understand that antibiotics do not work against colds and flus caused by a virus [7, 8]. This misunderstanding might be a reflection of antibiotics being prescribed and used for indistinguishable common conditions caused by either bacteria or viruses in Myanmar [28]. Altogether, these findings indicated knowledge gaps with respect to understanding antibiotics and resistance among the general public, in addition to reflecting intra-country differences between urban and rural residents, and younger and older citizens. As such, the intensive, nationwide dissemination of knowledge is required in Myanmar; this could be accomplished by developing and delivering information, education, and communication materials via multiple communication channels tailored for different targets.

Approximately half of the participants who recognized antibiotics had previously used antibiotics themselves (54.7%); this percentage is considerably lower than that noted via surveys in neighboring countries (>90%) [7, 8]. Noticeably, people in Myanmar were less likely to have

recently taken antibiotics, with only 10% of participants having taken antibiotics within the past month. This proportion is similar to that noted in Thailand (8%), but is considerably lower than those noted in surveys conducted in India (53%), Vietnam (42%), and Indonesia (34%) [7, 8]. Moreover, in the previous six months, only 31% of survey participants from Myanmar reported taking antibiotics, whereas a considerably higher percentage was found among survey participants from India (84%), Vietnam (78%), and Indonesia (74%). In addition, according to a recent study of antibiotic consumption in Myanmar, consumption in public hospitals has decreased over time [28]. At present, the overuse of antibiotics does not appear to be a significant problem in Myanmar, which could positively contribute to containing antibiotic resistance.

Most participants acquired their latest antibiotics from a clinic, medical store, or local pharmacy. This finding is relatively consistent with those of surveys from neighboring countries like India (95%), Vietnam (95%), Indonesia (96%) and Thailand (98%) [7, 8]; however, compared with those neighboring countries, a larger proportion of individuals acquired antibiotics from non-qualified providers in Myanmar, such as stalls, hawkers, friends, or family members. In addition, approximately two-thirds of individuals acquired antibiotics without a medical professional-issued prescription, consequently accelerating the misuse of antibiotics in Myanmar. Although the Drug Law in Myanmar stipulates that antibiotics can only be prescribed by qualified medical personnel, in reality, antibiotics are distributed in the general market and can be purchased by anyone, without a prescription [29–31]. Furthermore, other studies reported that substandard and falsified antibiotics can be found in general markets in developing countries [32–35], and that the consumers who purchase from unauthorized sales outlets in the market are at high risk of receiving substandard antibiotics [34]. If this situation continues to go unmonitored, increased inappropriate antibiotic use would further increase the problem of antibiotic resistance; to prevent this issue from becoming larger, existing drug law must be reinforced and strengthened.

In rural areas, where antibiotics were more frequently obtained without a prescription, the insufficient number of medical facilities and medical personnel might result in a situation in which obtaining antibiotics may entirely depend on the discretion of non-medical staff at the community pharmacy or kiosk, suggesting that community pharmacists and shop staff may play a critical role in facilitating antibiotic stewardship in rural areas [36–38]. As such, in addition to medical professionals, these non-medical key stakeholders should also be involved in the national antibiotic stewardship program.

Another noteworthy stewardship-related finding was that a larger proportion of individuals in Myanmar stopped antibiotics when they felt better, compared to those in neighboring countries like India (37%), Vietnam (38%), Indonesia (22%) and Thailand (28%), where most people reported completing the course of their antibiotic treatment as directed by health professionals [7, 8]. Self-medication without medical professional consultation may be attributed to multiple factors, such as the individuals assuming that the illness was a minor health issue, a desire to avoid the high costs and inconvenience associated with visiting a medical professional, having past experience with a similar illness, believing that antibiotics can accelerate recovery for all illnesses or prevent other infections, and not being aware of the side effects of antibiotics [39, 40]. This practice may also be a risk factor to inducing additional antibiotic resistance in Myanmar. As such, prescriptions from medical professionals, based on their medical assessment and advice to complete the course of all prescribed antibiotics, are the key to promoting the appropriate acquisition and use of antibiotics.

Approximately half of all participants correctly understood that the unnecessary or inappropriate use of antibiotics could result in ineffective treatment or resistance and had heard about antibiotic resistance in Myanmar; meanwhile, a higher awareness level was observed in

neighboring countries, like India (76%), Vietnam (78%), and Indonesia (84%) [7]. In Myanmar, more effort is needed to increase the public's awareness of antibiotic resistance. With respect to the sources of information on antibiotic resistance, most participants received information from medical professionals and family or friends, whereas TV and Facebook were the primary information sources in terms of media. As planned in the national AMR action plan of 2017–2021 in Myanmar, effective communication and awareness campaigns on antibiotic resistance should be immediately implemented using effective means of reaching individuals who are unaware of and have misconceptions about antibiotics, antibiotic use, and antibiotic resistance. Communication through medical professionals, such as doctors and nurses, may be one of the best options to promote better understanding and awareness among individuals in Myanmar [41]. Given that family members and friends also acted as vital communication hubs, workplace, school, and community campaigns might also be effective. In terms of media, TV and Facebook should also be utilized to effectively disseminate the campaign's messages [42]. Since the media played a critical role in enabling the people to have a greater chance of exposure to health and medical information during the COVID-19 pandemic [43, 44], those media could be utilized effectively to improve the public's knowledge and practices with respect to antibiotics.

In this study, the participants were less likely to understand that antibiotics are used for sick farm animals although it was better than the people from Thailand (24%). In comparison, more than two-thirds of survey participants in India and Vietnam were aware of this fact [7, 8]. Furthermore, in Myanmar, more than two-thirds of survey participants did not know that using antibiotics in animals could develop antibiotic resistance, and that the government had banned the use of antibiotics as a growth stimulator in livestock. The overuse and misuse of antibiotics in animals also accelerates the challenge of antibiotic resistance in humans. The survey also highlighted a significant gap in the knowledge and awareness of antibiotic use in farm animals and livestock. The national AMR program should further collaborate with the agriculture, livestock and veterinary department for providing additional opportunities for farmers to acquire trainings, education and information on antibiotic use and resistance in farm animals.

There are several limitations to this survey. First, this was an MPPS which might affect selection bias, especially sampling and non-response bias. The survey did not include individuals who do not possess mobile phones, as well as individuals from two regions and states with low population numbers. Furthermore, the high non-response rate included only individuals who accepted participating in the survey. However, advantageously, the MPPS approach allows for the quick and repeated collection of information from the same participants in broader areas and replacing those non-responses with different people with similar characteristics immediately, with lower costs than a traditional household survey [18, 45, 46]. Since this awareness survey must be conducted repeatedly, as planned in the National Action Plan [13], to track the impact of awareness-raising efforts, to ensure that campaigns targeting the public address key knowledge gaps and correct common misunderstandings, the MPPS is the most feasible and cost-effective data collection methodology for this survey. Second, the number of participants was insufficient for the required sample size due to excluding missing data and those that did not fit the inclusion criteria, although we collected more data than required. However, weighting the data based on the national census would complement the insufficiency and bring them closer to the national representativeness.

Despite the limitations mentioned above, this study is the first large-scale survey to cover a broader range of populations, both in terms of age and geography, and measure the level of public knowledge and awareness of antibiotics and antibiotic resistance in Myanmar. Since this survey is one of the activities planned in the National Action Plan for Containment Antimicrobial Resistance (2017–2022), the survey results must facilitate practical actions toward

ensuring the appropriate antibiotic use among the general population through effective public education and awareness campaigns.

## Conclusion

This survey provided baseline evidence on public knowledge and awareness of antibiotics and antibiotic resistance in Myanmar prior to the COVID-19 pandemic. Our results highlight large gaps in the knowledge on antibiotics and antibiotic use, awareness of antibiotic resistance, and antibiotic acquisition and use practices in Myanmar, compared to other countries. An understanding of the general public's knowledge and awareness of antibiotics and antibiotic resistance is vital to ensuring that the national program implements a technically-sound intervention for controlling AMR. This awareness survey should be conducted regularly to assess awareness level trends among the general population, as well as to evaluate the effectiveness of any implemented interventions. An awareness survey among different populations, such as medical professionals and pharmacies, and among people with different socio-economic characteristics, such as education, income, and employment, also requires to be done to highlight specific actions that should be taken with respect to tailored approaches to this population of individuals. Thus, the findings in this survey should be incorporated into the new national AMR strategy and its action plan and utilized for planned public campaigns to raise public awareness and improve antibiotic stewardship in Myanmar.

## Supporting information

**S1 File. Survey questionnaire (English).**
(DOCX)

## Acknowledgments

We would like to express our sincere thanks to the persons for their voluntary participation in the survey. We greatly acknowledge all of the surveyors and supervisors from the Myanmar Survey Research, and the microbiology department staff from the National Health Laboratory, Myanmar.

## Author Contributions

**Conceptualization:** Shinsuke Miyano, Thi Thi Htoon, Ikuma Nozaki.

**Formal analysis:** Shinsuke Miyano, Thi Thi Htoon.

**Funding acquisition:** Shinsuke Miyano.

**Methodology:** Shinsuke Miyano.

**Supervision:** Thi Thi Htoon, Ikuma Nozaki, Eh Htoo Pe, Htay Htay Tin.

**Validation:** Eh Htoo Pe, Htay Htay Tin.

**Writing – original draft:** Shinsuke Miyano.

**Writing – review & editing:** Thi Thi Htoon, Ikuma Nozaki, Eh Htoo Pe, Htay Htay Tin.

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
