## [Decision Letter · Decision Letter 0]

25 Apr 2022

PONE-D-22-06745Public knowledge, practices, and awareness of antimicrobials and antimicrobial resistance in Myanmar: The first nationwide mobile phone panel surveyPLOS ONE

Dear Dr. Miyano,

Thank you for submitting your manuscript to PLOS ONE. After careful consideration, we feel that it has merit but does not fully meet PLOS ONE’s publication criteria as it currently stands. Therefore, we invite you to submit a revised version of the manuscript that addresses the points raised during the review process.

This investigation has some value and the findings are worthy of broader dissemination. However, reviewers identified several key methodological problems and inconsistencies throughout the paper, recommending that it be thoroughly edited and updated for improved clarity and consistency. The methodology section, in particular, needs to be fleshed out further than it is now.

We look forward to receiving your revised manuscript.

Kind regards,

Myo Minn Oo, M.D., Ph.D.

Academic Editor

PLOS ONE

Journal Requirements:

Reviewers' comments:

Reviewer's Responses to Questions

**Comments to the Author**

1. Is the manuscript technically sound, and do the data support the conclusions?

Reviewer #1: Yes

Reviewer #2: Yes

Reviewer #3: Partly

Reviewer #4: Yes

Reviewer #5: Partly

Reviewer #6: Yes

Reviewer #7: Yes

Reviewer #8: Yes

2. Has the statistical analysis been performed appropriately and rigorously? 

Reviewer #1: Yes

Reviewer #2: Yes

Reviewer #3: No

Reviewer #4: Yes

Reviewer #5: No

Reviewer #6: No

Reviewer #7: Yes

Reviewer #8: N/A

3. Have the authors made all data underlying the findings in their manuscript fully available?

Reviewer #1: No

Reviewer #2: Yes

Reviewer #3: No

Reviewer #4: Yes

Reviewer #5: Yes

Reviewer #6: No

Reviewer #7: Yes

Reviewer #8: No

4. Is the manuscript presented in an intelligible fashion and written in standard English?

Reviewer #1: Yes

Reviewer #2: Yes

Reviewer #3: Yes

Reviewer #4: Yes

Reviewer #5: Yes

Reviewer #6: Yes

Reviewer #7: Yes

Reviewer #8: No

5. Review Comments to the Author

Reviewer #1: Abstract:

- Better to mention the survey period

Introduction:

- Better to show some highlights from AMR awareness surveys (India, Indonesia, and Thailand) of WHO South-East-Asian region

- It will be better if the author can brief a paragraph about the highlights on magnitude and prevalence of AMR in Myanmar if the data is available

Materials and Methods:

- It is needed to describe a brief on the sampling methodology (how to get the sampling frame and how the sample from the sampling frame were selected)

General Comments on Tables

- Tables and figures should be standalone and contains as much information related to the data as possible and abbreviations used in the tables should be described in long form in foot notes.

- It is better to highlight/bold/annotate the significant p-values for better visualization

Table 2:

- Title: to include the where and when of the survey

- To add footnote and should mention what is the % means (e.g., row percentage or column percentage)

Table 3:

- Same comment as table 2

Table 4:

- Same comment as table 2

Table 5:

- Same comment as table 2

Table 6:

- Same comment as table 2

Discussion:

- It is better to add a paragraph on strength of this study

Reviewer #2: 1. In the data collection method, please mention how many trained interviewers were used in this study and add the average duration of the phone interviews.

2. In Table (1), A3 question stated that “If yes in A2, where did you acquire the last course of antibiotic?”, but the above A2 question was not “yes/no” question. Therefore, please make sure A2 question is correct.

Reviewer #3: 

I am thankful to the editors and authors for giving me a chance to review this manuscript. It's overall well written by the results from a mobile phone panel survey among relatively large samples of the Myanmar population. For me, however, major improvements should be made on materials and methods and result sections to publish it in PlosOne. 

Abstract

- Page 2, Line 33, the word “younger individuals” is misleading. Consider rephrasing or including the age range in years. 

- Page 3, Line 39, a concluded sentence is unclear for me. What does it mean to “effective interventions”? 

Introduction

- Introduction is relatively short and lacks detailed information concerning the AMR situation in Myanmar, the current healthcare system including challenges towards universal health coverage, and weaknesses in drug regulations, etc. 

- It’d be better to include why authors chose the mobile phone panel for data collection and its efficiency referencing other literature or studies. 

Materials and Methods

- Authors used MPPS as a data collection tool in this study. However, I believe authors should describe it in detail so that readers can understand easily about it.

- By a mobile phone panel, how did you ensure the involvement of similar gender, age, and residential area? 

- Please include operational definitions of the variables (e.g., what does it mean “Urbanity”?)

- For data collection, given that mobile phone connections are a bit terrible in Myanmar, there might be some dropped out calls. How did you handle it? 

- There might also be some people holding multiple numbers.

- We should have included routine QA and QC for collected data, if any.  

- How did you validate the age of respondents by a call. Other family members could pick up phone calls rather than the owner of your intended number. Amid an increase in the number of dialers by some companies to promote their products or services, people nowadays avoid picking up such calls.

- How did you verify the translation of the questionnaire is correct and concise (e.g., back-translation method)?

- Authors mentioned making modifications to some questions after the pretest. What were those? Did you run any statistics to test its validity and reliability? Please consider attaching the Questionnaire in Burmese as a support file. 

- Questionnaire was composed of four sections. How about general characteristics such as age, gender, and residential areas of respondents?

-  Describes full name when it appeared first (e.g., SPSS)

- Line 117 – Authors mentioned about 95% CIs, but I did not find it in the later part. 

- Please refer to the correct name of the IRB board from Myanmar @ https://irbdmr.gov.mm/.

Results

- Analysis is a kind of thin. I have observed that there were inequalities in the residential areas of the respondents. How did the authors make an account of such bias? 

- In Table 1, there were some open-end questions (e.g., Q-C4). However, I did not find detailed results under the result section. 

- Please include footnotes under each table as relevant (e.g., degree of freedom for Chi-square test, differentiations whether Fisher Exact or Chi-square is used for each variable) 

 - Age range (mean, min and max) for the respondents should be included. Some older people might have a problem in answering the phone or they are dependent on other family members who are usually taking care for their medications. 

Discussion

- This part looks vast by including many things from the result section. I would suggest revising it in concise form, consider adding more about authors insights reflecting current situation in Myanmar. 

- One other limitations of this study would be lacking information from provider side (e.g., Knowledge of pharmacist/ pharmacy). Normal person might be remote from A/ AMR awareness, at least, most respondents went to clinics, medical stores, or pharmacies for buying medicines rather than seeking self-treatment.    

- Line 306-307, consider rephrasing the sentence to reflect current COVID-19 preventive measures in Myanmar. 

Other minor comments

- Please recheck the format of the references, e.g., 1, 3, 7, 12, 14, 15. 

- Ensure language errors, e.g., line 85 “High-frequency panel data nationwide.”  

Reviewer #4: Reviewer’s comment

Overall comment:

The manuscript tackles an important research area, antimicrobial resistance, which is a global health concern. The findings of the study were applicable as it identified the gaps in the knowledge, practices and awareness on antimicrobial resistance among Myanma population. The survey method was Mobile Phone Panel Survey approach which was reported as a successful cost-effective tool during these years. However, there are some major concerns regarding research tool and some facts that the authors may need to clarify for the improvement of the manuscript.

Major Comments

- Regarding the title, the authors may need to take a care while using the word “nationwide” . That issue is needed to be clarify in this revision. So after clarifying this, the title should be considered again.

-

- Regarding the methodology section, it is need to clarify the followings;

- how to achieve the national representative sample for Myanmar? Which sampling method was used to achieve representative MPPS for nationwide distribution?

- The author described the sample size but may need to mention the sampling frame clearly.

- How the authors perform representative MPPS?, face-to-face baseline interview approach was not conducted? Which approach was used to get the phone numbers of potential respondents/participants? random digit dialing (RDD) ? How to get a database of telephone numbers to draw a representative sample?

- how did you select the participants from each regions and states in Myanmar to get representative samples for urban/rural population within the each state and region?

- The authors described that out of 14 states and regions, participants from 12 states and regions were included; the author may need to mention which state/region were not included in the study and what is the reason (I found low population area in the discussion section) but need to clarify clearly? and need to mention how many representative participants were included in each state and region as the distribution of population is different among states and region in Myanmar. Were urban and rural population samples distributed equally within the each state/region or within the whole country?

- The authors mentioned in the Data Analysis session about the weighing factor was calculated based on National Census Data for gender/age/urbanity, please describe details how to stratify these three groups and how to ensure the validity of these info of the participants?

- The classification for “rural and urban area” is needed to be mention, which references ? census data? Moreover, a working definition for “residence of rural and urban area “ should be included eg. The respondence must be at least --- years of residing in rural/urban area etc.., Did the authors exclude those who just move in in urban area?

- Regarding the Reference section, many references were found to be following the Journal’s referencing style, the authors should take a thorough care to follow strictly to the author’s guidelines of the Journal

Minor comments

Line 20- pls correct as “Myanmar National Action Plan for Containment of Antimicrobial Resistance (2017-2022)”

Line 21- Pls write as “Ministry of Health and Sports, Myanmar”

Line 24- -Pls add the time of the study; eg.“….conducted MPPS during 2020.. ”

Line 33- Pls mention the age group instead of “younger”

Line 112- Pls mention “SPSS” in full “Statistical Package…..”

Line 136- table 2 should be after the paragraph for “Recognition of antibiotic”, is the recognition used there is the correct term?

Pls delete “full stops” in the titles of table 2,3,4 and 5

Line 230-232 is repeating the Line 227-229

Line 322 -333- It is more suitable in introduction section as it explained about MPPS and Myanmar situation. In the discussion the author should mention limitation (as in Line 320-322) and advantages (as in Line 333-338) of the present study precisely.

Line 374- Strep pneumoniae- italic

Ref 7 and 14 are overlapping, and the ref style for WHO, European Union (Ref 15) were not correct and other references are needed to be checked with Journal’s referencing style

Reviewer #5: This study is important for baseline data for the antimicrobial use and antimicrobial resistance in Myanmar. As this study is the first nationwide data, there are some considerations especially methodology I want to point out.

Title: Public knowledge, practices, and awareness of antimicrobials and antimicrobial resistance in Myanmar: The first nationwide mobile phone panel survey

But the authors stated their aim of the study as "this study aimed to assess public knowledge and practices concerning antibiotics, as well as awareness of antibiotic resistance, to provide baseline evidence and identify gaps related to the deployment of strategic awareness-raising activities in Myanmar". Moreover, the questionnaires are asked about antibiotics only.

For non -expert readers, I think some clarifications are needed in the terms of antimicrobial or antibiotics (or) changing the title by using antibiotics only

Introduction

The authors should include other previous evidences from Myanmar even though they are not nationwide. From that, international audience can estimate the prevalence and burden of the interesting issue in Myanmar and the authors can explain the results in discussion.

Method

Before going to details, the authors should follow STROBE (for cross sectional study) reporting guideline especially for the method section because this is the most important part to decide the validity and reliability of the study.

Sample size:

line 85 to 94: The authors stated the steps to get the required sample size 1940. But there is difference what the reference book stated. According to the reference, Mobile phone panel survey in developing countries by world bank, in page 24, sample size calculation https://openknowledge.worldbank.org/bitstream/handle/10986/24595/9781464809040.pdf?sequence=2&isAllowed=y, the required sample size will be 3876 for 6 layers of stratifications (urban/rural, male/female, Younger/older). That is only for 2 layers of age stratification. If you have more layers, it will be increased according to the layers.

Please refer to the above link and justify the required sample (or) only mention another appropriate sample size calculation techniques.

Questionnaires:

Line 101 to 104: The authors stated the pretest was done. If any item analysis was done, please stated the results which will increase the validity of the questionnaires. Moreover, the study was claimed as nationwide study and to get baseline information, the authors should express detail procedure of development of the questionnaires and testing the internal and external consistency. I

Usually, the questionnaires contain general information of the study population as part A. I believe the authors already had that part. If so, please mention it and the data must show in the result section also. Because the readers want to know the general information of the study population.

According to the question A1, the population will divide 2, Yes and no. Further questions are based on "Yes" sub population. It is not clear whether the authors apply questions B, C and D to all participants or only to those who answer to Yes in question A1. Please mention it clearly in the method section and data analysis section for any sub group analysis.

Data Analysis: Thank to the authors for considering the sampling weights to reduce the bias. But need to mention any sub group analysis and how to handle the nonresponse rate because non response rate is important for MPPS.

Since the authors expressed the difference between groups, I suggest to calculate multivariable analysis for confounder-adjusted estimates to minimize the bias and show the strength of association.

Result

Again, I recommend to stick the reporting guideline. Please give the characteristics of study participants like Age, gender, residence, education, occupation, socioeconomic status etc

Please indicate number of participants with missing data or non-response rate. Since this is a survey type study, the readers want to know the non-response rate.

I know the table 3,4 and 5 are representing the sub groups. I am not sure the table 6. Please specify the total number of each group in the heading of the tables and in the test also. It is confused the readers. It is better if the authors show the flow diagram of the number of the response of the participants.

Discussion

The whole discussion is well written and can explain the result well. Recommendation is also appropriate. But it can change after the authors adjust the data analysis. As I mention before, more Myanmar contexts/studies should be discussed here.

Strength and limitation of the study

Line 325-338, the authors want to express why they use MPPS. So, the whole para graph should express in the introduction or beginning of the method.

The authors should address other important factors such as social and economic factors which influence the results. Any potential bias should be expressed here.

Reviewer #6: The study covers important public health topic and it is very timely. The manuscript reads very well.

Some comments for the better understanding on the method applied and findings.

Method

Line number - 84-85/ 98 - Please consider giving some clarification on the phrase "representative, high-frequency panel data". And some detail on how the sample frame was identified? How the samples were randomly selected?

Was response rate taken into account when sample size was calculated?

Line number - 91 - there was a plan to do series of survey and sample size was calculated to compensate 5% attrition rate. It is suggested to elaborate more about the plan in discussion or where relevant.

Line 102 - Any references used in the development of the questionnaire? Did the authors use standard questionnaire and any reference on its reliability?

Line 111 - Data were double-entered into Excel. Please include the reason of doing double entries and was there any inconsistency when/if the authors did data validation? If so, how were the errors managed?

Line 117 - Method used for calculating 95% CI was presented but no 95% CI was reported in the result session. It is suggested to include 95% CI.

Important socio-economic factors such as education, income, occupation should be considered including in future surveys.

Results

What is the response rate?

Line 133-134 - 68.5% lived in urban area. It is different from the data presented in table 2 where 68.5% lived in rural.

When giving the p-values for difference groups comparison in result narrative, it is better to move the p value to the place near to variable name rather them writing all p values in the parenthesis at the end of the statement which sometimes is difficult to interpret.

Conclusion

Line 349-351 - Prediction on the use of antibiotics during the COVID19 pandemic is not based on the current study findings and better to rephrase as a recommendation and/ or move it to discussion by giving the appropriate references.

Reviewer #7: Congratulations for your well prepared manuscript.

May I suggest to add some modifications by the following comments:

Line 22-23: Paraphrase of the aim will be better if it is consistent with the title of manuscript.

Line 93: Reference for sample size consideration and formula for attrition rate should be cited.

Line 99: How are the panels framed for random selection of each participant?

Line 316: Awareness raising on antibiotic use, antibiotic resistance and banning of antibiotic use in farm animals should be highlighted for rural people because they occupied for about two third of Myanmar population and are exposed more to farm animals than urban residents.

Line 306-307, 332-333: Along with increasing ownership of mobile phones in Myanmar, mobile health education by mean of text messages about antimicrobials use and antimicrobial resistance should be recommended as communication channel of awareness raising campaign.

Reviewer #8: 

Summary of the research and overall impression This manuscript presented to access the knowledge, awareness and practice of using antibiotics among general populations in Myanmar using MPPS. Methodology session is required to revise/ add more detail information in terms of sampling, study period, data collection approach and data management. Data analysis is stated clearly. The results session is clear but the discussion and conclusions are required to improved. This study will benefit to the community and national AMR program to strategically implement the national AMR action plan. I would agree to publish this article for wider data dissemination purpose to the academic audience as well as useful for the uptake by the national program. It is recommended to improve the English language for academic writing and publication. For example, the use of tense (verb) in the results and discussion sessions should be past tense. The terms should be consistent: “participants” or “respondents” or “citizens” throughout the paper. Similarly, the term “overuse” or “misuse” should be consistent. If “overuse” and “misuse” are different definition, please elaborate in methods session.  Discussion on specific area of improvement Abstract1.
The information presented in the abstract is clear, concise and complete. Thank you. Please add how the participants were randomly selected from which sampling frame. It said “from the panel” – please elaborate more what is the “panel” represent for. Introduction 2.
Add the key findings from the 3 countries in SEA region (India, Indonesia and Thailand) – eg. the proportion of public awareness and understanding about AMR in these counties. 3.
List the countries where the WHO AMR nationwide survey was conducted in the SEA region. 4.
Mention additional information about the national AMR program in Myanmar and the action/ activities implementing for AMR strategic action plan. Methods5.
Explain the sampling procedure – how the participants were chosen from the panel data nationwide. 6.
Table 1 – please move it to Annex.  7.
Clarify the “period of the study” and “study areas” included in the study where the participants were enrolled in the survey.8.
Suggest to mention about “data management”, such as data cleaning and checking, how the data was stored for such a certain period of time, data security measure, etc.9.
Add the information about non-response rate, which is important for mobile phone interviews using IVR at the call center. 10.
Mention how the data quality check was conducted (in a real-time?) for IVR and mobile phone interviews. Any follow up questions if the data is incomplete or occurs with errors. Ethical considerations 11.
Line 126 – please double check the name of the institutional board for ethical review. I assume it should be “Institutional Review Board”, not “Institutional Research Board”. Results 12.
Move the study period to the methods session. 13.
The sample size (in the methods session) said 1940 participants, but how it was ended up with 2045 participants. Please explain in the methods session and in results session, total sample size is enough to present. 14.
Line 133 – 68% lived in RURAL areas (according to the data from the table). Please correct. 15.
Mention the proportion of participants in each state/ region in the Annex table. Discussion and conclusions16.
Line 226 – “excellent representation” is not the academic writing; reflecting cannot measure how much “excellent” stands for. Please mention scientifically for the representativeness of the sample populations from the study. 17.
In general, add the data from other literatures/ studies when explaining about the comparisons of data from this study findings, rather than mentioning “less likely” or “more likely”, so that the readers can get more detail understanding of how the differences is small or large in this study, compared to other studies. 18.
Line 242-245 (…nationwide dissemination……multiple communication channels.) – how did you get this recommendation to be accomplished the knowledge dissemination in Myanmar. Please refer to and double check with the findings that can reference for this recommendation. 19.
Line 266-267 (…in reality……without prescription.) – this statement is not included in the findings. Please add the reference if relevant or please remove the statement if that is not from the findings or no reference can be added. You may not want to add information without any evidence or reference in the discussion. 20.
Line 309 – you cannot state “people from Myanmar” because this study was conducted in a sample population of Myanmar. You can state that “in this study, participants were less likely….”21.
Line 317-319 – suggest to mention that further collaboration with the agriculture, livestock and veterinary department for providing proper training and education program to the farmers and livestock, breeding farms regarding the AMR. 22.
Limitation: mention the weakness of mobile phone survey, instead to face-to-face survey (eg. non-response rate). 23.
Please mention anything about the safeguarding measures applied throughout the MPPS survey approach. If not, add this as the limitation. 24.
However, the later part of the paragraph (line 327 onwards) stated about the advantage of MPPS. In this case, please present clearly “limitation” and “strength” of the survey. In addition, please also add the literature and reference about using MPPS methodology. Reference from Oxfam related to CATI (using phone interview) is provided for additional reference as necessary. Conclusion 25.
COVID-19 related AMR situation is just appeared in the conclusion. It is agreeable to include and discuss about this survey conducted before COVID-19 pandemic. Thus, include or move some information about COVID-19 related AMR situation in the discussion first, then present it again to conclude how this data could be impacted after the COVID-19 pandemic. 26.
The facts in conclusion should be more concisely presented one by one, rather than general presentation. **********

6. PLOS authors have the option to publish the peer review history of their article (what does this mean?). If published, this will include your full peer review and any attached files.

Reviewer #1: No

Reviewer #2: No

Reviewer #3: No

Reviewer #4: No

Reviewer #5: No

Reviewer #6: No

Reviewer #7: No

Reviewer #8: **Yes: **POE POE AUNG

---

## [Author Response · Author response to Decision Letter 0]

18 Jul 2022

Reviewer #1

Abstract:

- Better to mention the survey period

We appreciate the reviewer's comment on this point. We have added the survey period in the abstract (January and February 2020).

Introduction:

- Better to show some highlights from AMR awareness surveys (India, Indonesia, and Thailand) of WHO South-East-Asian region

We appreciate the reviewer's suggestion on this point. We have added brief highlights from AMR awareness surveys of WHO South-East-Asian region. Detail findings are also explained in the “Discussion” as a comparison of findings in between those countries and Myanmar.

- It will be better if the author can brief a paragraph about the highlights on magnitude and prevalence of AMR in Myanmar if the data is available

We appreciate the reviewer's suggestion on this point. We have added a prevalence of AMR in Myanmar (WHO priority pathogens) from the national AMR report published in 2019, highlighting the magnitude of AMR in Myanmar.

Materials and Methods:

- It is needed to describe a brief on the sampling methodology (how to get the sampling frame and how the sample from the sampling frame were selected)

We appreciate the reviewer's suggestion on this point. We have added the explanation on the sampling methodology. The panel is composed of 200,000 contacts that are updated regularly. Field workers across the country interview potential respondents in person at their homes, collect their basic demographic information, and receive their permission to be included in the panel to respond to several MPPSs on various topics. In this survey, the total planned sample of 2,126 was distributed equally across 16 demographic segments, which are gender (male and female), age group (18-29, 30-39, 40-49, and ≧50 years old), and residential area urbanity (urban and rural), and 133 samples were allocated in each segment for sampling. Those samples were selected randomly from the panel automatically.

General Comments on Tables

- Tables and figures should be standalone and contains as much information related to the data as possible and abbreviations used in the tables should be described in long form in foot notes.

- It is better to highlight/bold/annotate the significant p-values for better visualization

We appreciate the reviewer's comment on this point. We have annotated the significant p-values as recommended.

Table 2:

- Title: to include the where and when of the survey

We appreciate the reviewer's comment on this point. We have added the information (place and time) of the survey.

- To add footnote and should mention what is the % means (e.g., row percentage or column percentage)

We appreciate the reviewer's comment on this point. We have added the information on the % at the footnote.

Table 3:

- Same comment as table 2

We appreciate the reviewer's comment on this point. We have done the same as your suggestion for table 2.

Table 4:

- Same comment as table 2

We appreciate the reviewer's comment on this point. We have done the same as your suggestion for table 2.

Table 5:

- Same comment as table 2

We appreciate the reviewer's comment on this point. We have done the same as your suggestion for table 2.

Table 6:

- Same comment as table 2

We appreciate the reviewer's comment on this point. We have done the same as your suggestion for table 2.

Discussion:

- It is better to add a paragraph on strength of this study

We appreciate the reviewer's comment on this point. We have added a paragraph on strength of this study after the paragraph on limitation.

Reviewer #2

1. In the data collection method, please mention how many trained interviewers were used in this study and add the average duration of the phone interviews.

We appreciate the reviewer's comment on this point. We have added the information in the Method and Result section. Nineteen trained interviews were used in this survey, and the average duration of the phone interview was 7.53 minutes. 

2. In Table (1), A3 question stated that “If yes in A2, where did you acquire the last course of antibiotic?”, but the above A2 question was not “yes/no” question. Therefore, please make sure A2 question is correct.

We appreciate your pointing out our mistake. A2 question was not correct. We have corrected the question and attached the questionnaire both in English and Burmese as Annex 1.

Reviewer #3: 

I am thankful to the editors and authors for giving me a chance to review this manuscript. It's overall well written by the results from a mobile phone panel survey among relatively large samples of the Myanmar population. For me, however, major improvements should be made on materials and methods and result sections to publish it in PlosOne. 

Abstract

- Page 2, Line 33, the word “younger individuals” is misleading. Consider rephrasing or including the age range in years. 

We appreciate the reviewer's suggestion on this point. We have added the age range in years (18-29 years).

- Page 3, Line 39, a concluded sentence is unclear for me. What does it mean to “effective interventions”? 

We appreciate the reviewer's comment on this point and agree that “effective interventions” sounds unclear. We have revised it into “continuous public education and awareness campaigns”.

Introduction

- Introduction is relatively short and lacks detailed information concerning the AMR situation in Myanmar, the current healthcare system including challenges towards universal health coverage, and weaknesses in drug regulations, etc.

We appreciate the reviewer's suggestion on this point. We have added the information on the AMR situation in Myanmar, using the data of other studies as references. However, to focus on the AMR in this paper, we would like not to broaden topics like health systems in the “Introduction” of this manuscript. We hope the reviewer could understand our idea.

- It’d be better to include why authors chose the mobile phone panel for data collection and its efficiency referencing other literature or studies. 

We appreciate the reviewer's suggestion on this point. We have included why we chose the MPPS for data collection and its efficiency rather than household survey, using some references in the Introduction session. The advantages of mobile surveying are manifold: they allow respondents to be contacted remotely at a time of their convenience; provide timely and low-cost alternatives to traditional face-to-face survey administration; and permit results to be fed back to evaluators in near-real-time

Materials and Methods

- Authors used MPPS as a data collection tool in this study. However, I believe authors should describe it in detail so that readers can understand easily about it.

We appreciate the reviewer's suggestion on this point. We have added some explanation on the MPPS in the Method, including what the mobile phone panel we used were, how the panel was created and updated, and how we chose the participants from the panel.

- By a mobile phone panel, how did you ensure the involvement of similar gender, age, and residential area? 

We appreciate the reviewer's suggestion on this point. We have added the information of the mobile phone panel in the Methodology. The mobile phone panel is composed of 200,000 contacts that are updated regularly. Field workers across the country interview potential respondents in person at their homes, collect their basic demographic information and receive their permission to be included in the panel to respond to several MPPSs on various topics, which made us ensure the involvement of similar gender, age, and residential area. To improve representativeness of the general population, when calculating the percentage of respondents who provided a particular response to each question, the sampling design and sampling weight of each participant were considered to ensure representativeness and unbiased results. In terms of the sample weight, the weighting factor was calculated using population data for each gender, age group, and residential area urbanity based on the latest National Census, conducted in 2014.

- Please include operational definitions of the variables (e.g., what does it mean “Urbanity”?)

We appreciate the reviewer's comment on this point. Urban and Rural are classified as per the government definition of areas as Urban and Rural. In Myanmar, top-level administrative unit is States/Regions, second level is District and third-level is Township. A Township consists of further administrative units which are classified as Urban by the government and called ‘Wards’ or it (frequently simultaneously) contain administrative units which are classified as Rural called ‘Village tracts’. Urban is classified when respondent is living ward area while living in village is considered as rural. This is determined by the interviewer when recording the residential address of the respondent during the interview.

- For data collection, given that mobile phone connections are a bit terrible in Myanmar, there might be some dropped out calls. How did you handle it? 

We appreciate the reviewer's comment on this practical point. Calls that were dropped out or with low voice quality due to network during the interview, are called back again immediately after hanging up. If the call back still faced issues in terms of voice quality, the interviewer takes an appointment with the respondent for call back. After three attempts, the respondent might be dropped and replaced with a different respondent with similar demographics (since demographic quota controls are in place). We have added this point in the Methodology (data collection) and Results (non-response rate).

- There might also be some people holding multiple numbers.

We appreciate the reviewer's comment on this point. The database of telephone numbers is organized with the name of the respondents. Those who hold more than a number was called once with the first number and if that first number is not picked up, the second number was dialed.

- We should have included routine QA and QC for collected data, if any. 

We appreciate the reviewer's comment on this very important point. Data validation is done daily by starting from the second day of the interviews till the next two days of the interviews ending. Quality control team selects randomly 30% of achievements per interviewer and investigates either by listening to the recordings or calling the respondent for verification. We have added this point in the Methodology- Data collection.

- How did you validate the age of respondents by a call. Other family members could pick up phone calls rather than the owner of your intended number. Amid an increase in the number of dialers by some companies to promote their products or services, people nowadays avoid picking up such calls.

We appreciate the reviewer's comment on this point. Respondents are firstly asked how old they are, followed by a question of their birthday to confirm if respondents have turned the age they answered on their last birthday. If the intended number is not answered by the owner, interviewers asked when they can call back to talk with the owner of the number.

- How did you verify the translation of the questionnaire is correct and concise (e.g., back-translation method)?

We appreciate the reviewer's clarification on this point. Yes, we did back-translation method. We have added this point in the Method-Data collection.

- Authors mentioned making modifications to some questions after the pretest. What were those? Did you run any statistics to test its validity and reliability? Please consider attaching the Questionnaire in Burmese as a support file. 

We appreciate the reviewer's question on this point. We discussed which word was used for “antibiotics” as there is no specific word for antibiotics in Myanmar language. We did not run any statistics to test its validity and reliability as the questionnaire is already used globally in several countries. We have attached the questionnaire both in English and Burmese as a supporting file (Annex 1).

- Questionnaire was composed of four sections. How about general characteristics such as age, gender, and residential areas of respondents?

We appreciate the reviewer's question on this point. We removed the section on general characteristics from the questionnaire in the manuscript to highlight the main point of the questionnaire. However, as we mentioned in the Method, we collected the general characteristics such as age, birthday, gender, residential areas of respondents.

- Describes full name when it appeared first (e.g., SPSS)

We appreciate the reviewer's comment on this point. We have added Statistical Package for the Social Sciences for SPSS. Apart from SPSS, we described full spell of abbreviated word when it appeared first.

- Line 117 – Authors mentioned about 95% CIs, but I did not find it in the later part. 

We appreciate the reviewer’s comment and pointing out our error. We have added 95% CI on the estimated percentage (%) of each response both in the sentences and tables.

- Please refer to the correct name of the IRB board from Myanmar @ https://irbdmr.gov.mm/.

We appreciate the reviewer's suggestion. We corrected the name based on the link the reviewer shared.

Results

- Analysis is a kind of thin. I have observed that there were inequalities in the residential areas of the respondents. How did the authors make an account of such bias? 

We appreciate the reviewer's comments on this point. In terms of the residential areas, to reduce the selection bias, we have weighted the collected data, using the National Census 2014, to make them close to the representative data. It is not perfect to clear the bias of residential area, but it is only the scientific way to reduce the bias. We have mentioned this way in the Method and Result and the limitation in the Discussion. 

- In Table 1, there were some open-end questions (e.g., Q-C4). However, I did not find detailed results under the result section. 

We appreciate the reviewer's clarification on this point. For these open-end questions, we are writing another manuscript as originally planned, so those results were not included in the Result section. To avoid the confusion, we have removed Q-C4 in the manuscript.

- Please include footnotes under each table as relevant (e.g., degree of freedom for Chi-square test, differentiations whether Fisher Exact or Chi-square is used for each variable) 

We appreciate the reviewer's suggestion. We have revised the manuscript in those points in the Methodology accordingly. All the analysis were done with Chi-square test. Degree of freedom depends on the number of characteristics (a) and the number of answers (b) in the questionnaire (calculated as (a-1)*(b-1)). Instead of putting that information in footnotes under each table, we have added the information in the Methodology.

 - Age range (mean, min and max) for the respondents should be included. Some older people might have a problem in answering the phone or they are dependent on other family members who are usually taking care for their medications. 

We appreciate the reviewer's suggestion on this point. We have added age range (mean 39.4, range 18-80). When the respondent had a problem in answering the questions, the interviewer replaced with a different respondent with similar demographics in the panel.

Discussion

- This part looks vast by including many things from the result section. I would suggest revising it in concise form, consider adding more about authors insights reflecting current situation in Myanmar. 

We appreciate the reviewer's suggestion. We have revised some parts of the Discussion accordingly. However, the challenges are that there are very limited number of papers/reports from Myanmar, when we search them in the Pubmed and Google Scholoar, as pieces of evidence to back up our insights. Instead, we put the comparison of the results between Myanmar and other countries in the Discussion. We hope the reviewer could understand it.

- One other limitations of this study would be lacking information from provider side (e.g., Knowledge of pharmacist/ pharmacy). Normal person might be remote from A/ AMR awareness, at least, most respondents went to clinics, medical stores, or pharmacies for buying medicines rather than seeking self-treatment. 

We appreciate the reviewer's comment on this point. Lack of information from provider side is very important point, however, it is not the limitation of the survey. Instead, we have already put this point in the Discussion and Conclusion. We were planning to conduct the survey of the provider side, but we were not able to implement it due to the political unstable situation.

- Line 306-307, consider rephrasing the sentence to reflect current COVID-19 preventive measures in Myanmar. 

We appreciate the reviewer's comment on this point. We have rephrased the sentence and reflected the current COVID-19 preventive measure with some references.

Other minor comments

- Please recheck the format of the references, e.g., 1, 3, 7, 12, 14, 15. 

We appreciate the reviewer's comments. We have corrected those references.

- Ensure language errors, e.g., line 85 “High-frequency panel data nationwide.” 

We appreciate the reviewer's pointing out our error. We corrected the language.

Reviewer #4:

Overall comment:

The manuscript tackles an important research area, antimicrobial resistance, which is a global health concern. The findings of the study were applicable as it identified the gaps in the knowledge, practices and awareness on antimicrobial resistance among Myanma population. The survey method was Mobile Phone Panel Survey approach which was reported as a successful cost-effective tool during these years. However, there are some major concerns regarding research tool and some facts that the authors may need to clarify for the improvement of the manuscript.

Major Comments

- Regarding the title, the authors may need to take a care while using the word “nationwide” . That issue is needed to be clarify in this revision. So after clarifying this, the title should be considered again.

We appreciate the reviewer's suggestion on the title. We have agreed the point and changed from “nationwide” to “national” as some areas are not covered by the survey due to some reasons written as the limitation.

- Regarding the methodology section, it is need to clarify the followings;

- how to achieve the national representative sample for Myanmar? Which sampling method was used to achieve representative MPPS for nationwide distribution?

We appreciate the reviewer's question on this point. We have used random sampling from the mobile phone panel. The mobile phone panel is an access panel, and the sampling method is analogous to online panels. In order to achieve a national representative sample, characteristics such as region/states, gender, age, and urbanity are considered. Sampling quotas are allocated in line with population characteristics (from 2014 Myanmar Census) to have a representative sample and the contacts matching the required quota for each target group are selected from the database. Quota control is essential to achieve a representative sample, as response rates to telephone surveys varies by demographics and more effort (i.e., more sample to be called) would be required to achieve certain quotas.

- The author described the sample size but may need to mention the sampling frame clearly.

We appreciate the reviewer's suggestion on this point. We have added this point in the Method.

The mobile phone panel is composed of 200,000 contacts that are updated regularly. Field workers across the country interview potential respondents in person at their homes, collect their basic demographic information and receive their permission to be included in the panel to respond to several MPPSs on various topics. In this survey, the total planned sample of 2,216 was distributed equally across 16 demographic segments, which are gender (male and female), age group (18-29, 30-39, 40-49, and ≧50 years old), and residential area urbanity (urban and rural), and 133 samples were allocated in each segment for sampling.

- How the authors perform representative MPPS?, face-to-face baseline interview approach was not conducted? Which approach was used to get the phone numbers of potential respondents/participants? random digit dialing (RDD) ? How to get a database of telephone numbers to draw a representative sample?

We appreciate the reviewer's clarification on this point. We have added this point in the Method.

The mobile phone panel is composed of 200,000 contacts that are updated regularly. Field workers across the country interview potential respondents in person at their homes, collect their basic demographic information and receive their permission to be included in the panel to respond to several MPPSs on various topics. In this survey, the total planned sample of 2,216 was distributed equally across 16 demographic segments, which are gender (male and female), age group (18-29, 30-39, 40-49, and ≧50 years old), and residential area urbanity (urban and rural), and 133 samples were allocated in each segment for random sampling.

- how did you select the participants from each region and state in Myanmar to get representative samples for urban/rural population within each state and region?

We appreciate the reviewer's clarification on this point. We selected the participants from each region and state in Myanmar based on probability proportional to size (PPS) sampling to get representative samples for urban/rural population within each state and region, using the latest public data from 2014 Myanmar Census.

- The authors described that out of 15 states and regions, participants from 12 states and regions were included; the author may need to mention which state/region were not included in the study and what is the reason (I found low population area in the discussion section) but need to clarify clearly? and need to mention how many representative participants were included in each state and region as the distribution of population is different among states and region in Myanmar. Were urban and rural population samples distributed equally within the each state/region or within the whole country?

We appreciate the reviewer's clarification on this point. We added the information/table on selected states and regions in the Result. Among 15 states and region, 2 states and regions (Chin, and Kayah) were excluded as the panel has relatively low representation for these states. Kayah and Chin represent 0.57% and 0.95% of national population and are relatively not significant in terms of sample allocation. Due to long periods or civil unrest and conflict, Rakhine State has been a problem for all survey agencies to conduct any fieldwork in, and this has also reflected in the composition of the mobile panel which has relatively low number of panelists from this state. Urban and rural population samples were distributed equally within the whole country.

- The authors mentioned in the Data Analysis session about the weighing factor was calculated based on National Census Data for gender/age/urbanity, please describe details how to stratify these three groups and how to ensure the validity of these info of the participants?

For this study, the total planned sample of n=2,216 respondents was spread equally across 16 ‘cells’ each representing a demographic segment for which it was important to have a robust base for analysis. The table is shown below (there are minor over-sampling done during fieldwork in order to match all quotas set).

 SAMPLE

Age group Male Female Total

 urban rural urban rural 

18- 29 6.4% 6.1% 6.1% 6.7% 25%

30- 39 6.1% 6.1% 7.0% 6.1% 25%

40-49 6.1% 6.6% 6.1% 6.1% 25%

50- 6.1% 6.1% 6.1% 6.1% 24%

Total 25% 25% 25% 25% 100%

The above numbers are not aligned with the actual Myanmar population distribution across the demographics of Gender, Age and Urbanity. The actual distribution is shown below:

 ACTUAL

Age group Male Female Total

 urban rural urban rural 

18- 29 5.3% 10.7% 5.6% 11.6% 33%

30- 39 3.6% 8.0% 3.9% 8.6% 24%

40-49 2.9% 6.5% 3.4% 7.2% 20%

50-60 3.0% 7.4% 3.9% 8.4% 23%

Total 15% 33% 17% 36% 100%

In order to correct the sample distribution and align it with Myanmar population distribution, weighting factors were calculated by dividing the proportion in the “Actual” table by the proportion in the “sample” table. This provided the weighting factors 

 WEIGHTING FACTOR

Age group Male Female

 urban rural urban rural

18- 29 0.8346 1.7577 0.9183 1.7282

30- 39 0.5863 1.3025 0.5553 1.4019

40-49 0.4675 0.9989 0.5554 1.1836

50-60 0.4959 1.2149 0.6301 1.3708

- The classification for “rural and urban area” is needed to be mention, which references ? census data? Moreover, a working definition for “residence of rural and urban area “ should be included eg. The respondence must be at least --- years of residing in rural/urban area etc.., Did the authors exclude those who just move in in urban area?

We appreciate the reviewer's comment on this point. Urban and Rural are classified as per the government definition of areas as Urban and Rural. In Myanmar, top-level administrative unit is States/Regions, second-level is District and third-level is Township. A Township consists of further administrative units which are classified as Urban by the government and called ‘Wards’ or it (frequently simultaneously) contain administrative units which are classified as Rural called ‘Village tracts’. Urban is classified when respondent is living ward area while living in village is considered as rural. This is determined by the interviewer when recording the residential address of the respondent during the interview. We did not collect the information the length of residing in rural/urban area that did not enable to exclude those who just moved in urban area.

- Regarding the Reference section, many references were found to be following the Journal’s referencing style, the authors should take a thorough care to follow strictly to the author’s guidelines of the Journal

We appreciate the reviewer's comment. We have corrected the referencing style as per the Journal’s guidance.

Minor comments

Line 20- pls correct as “Myanmar National Action Plan for Containment of Antimicrobial Resistance (2017-2022)”

We appreciate the reviewer's suggestion on this point. Accordingly, we corrected the name of the national document.

Line 21- Pls write as “Ministry of Health and Sports, Myanmar”

We appreciate the reviewer's suggestion on this point. We have corrected the name as suggested.

Line 24- -Pls add the time of the study; eg.“….conducted MPPS during 2020.. ”

We appreciate the reviewer's suggestion on this point. We have added the time of the study.

Line 33- Pls mention the age group instead of “younger”

We appreciate the reviewer's suggestion on this point. We have added the information on exact age group.

Line 112- Pls mention “SPSS” in full “Statistical Package…..”

We appreciate the reviewer's suggestion on this point. We have added the full name as suggested.

Line 136- table 2 should be after the paragraph for “Recognition of antibiotic”, is the recognition used there is the correct term?

We appreciate the reviewer's clarification on this point. “Recognition of antibiotics” means that they knew antibiotics that was asked in Question A1.

Pls delete “full stops” in the titles of table 2,3,4 and 5

We appreciate the reviewer's suggestion on this point. We have deleted them as suggested.

Line 230-232 is repeating the Line 227-229

We appreciate the reviewer's clarification on this point. Line 227-229 is the overall interpretation of the five true/false questions, but Line 230-232 is the interpretation of two exact questions among five questions. They are demonstrating the different points.

Line 322 -333- It is more suitable in introduction section as it explained about MPPS and Myanmar situation. In the discussion the author should mention limitation (as in Line 320-322) and advantages (as in Line 333-338) of the present study precisely.

We appreciate and agree on the reviewer's suggestion on this point. We have revised as suggested.

Line 374- Strep pneumoniae- italic

We appreciate the reviewer's suggestion on this point. We have corrected it accordingly.

Ref 7 and 14 are overlapping, and the ref style for WHO, European Union (Ref 15) were not correct and other references are needed to be checked with Journal’s referencing style

We appreciate the reviewer's pointing out our error. We have corrected the referencing style as per the Journal’s guidance.

Reviewer #5:

This study is important for baseline data for the antimicrobial use and antimicrobial resistance in Myanmar. As this study is the first nationwide data, there are some considerations especially methodology I want to point out.

Title: Public knowledge, practices, and awareness of antimicrobials and antimicrobial resistance in Myanmar: The first nationwide mobile phone panel survey

But the authors stated their aim of the study as "this study aimed to assess public knowledge and practices concerning antibiotics, as well as awareness of antibiotic resistance, to provide baseline evidence and identify gaps related to the deployment of strategic awareness-raising activities in Myanmar". Moreover, the questionnaires are asked about antibiotics only.

For non -expert readers, I think some clarifications are needed in the terms of antimicrobial or antibiotics (or) changing the title by using antibiotics only

We appreciate the reviewer's suggestion on this point. We have changed antimicrobials into antibiotics that reflected our survey more correctly in the title.

Introduction

The authors should include other previous evidences from Myanmar even though they are not nationwide. From that, international audience can estimate the prevalence and burden of the interesting issue in Myanmar and the authors can explain the results in discussion.

We appreciate and agree on the reviewer's suggestion on this point. We have added a prevalence of AMR in Myanmar (WHO priority pathogens) from the national AMR report published in 2019, highlighting the magnitude of AMR in Myanmar.

Method

Before going to details, the authors should follow STROBE (for cross sectional study) reporting guideline especially for the method section because this is the most important part to decide the validity and reliability of the study.

(Sample size)

line 85 to 94: The authors stated the steps to get the required sample size 1940. But there is difference what the reference book stated. According to the reference, Mobile phone panel survey in developing countries by world bank, in page 24, sample size calculation https://openknowledge.worldbank.org/bitstream/handle/10986/24595/9781464809040.pdf?sequence=2&isAllowed=y, the required sample size will be 3876 for 6 layers of stratifications (urban/rural, male/female, Younger/older). That is only for 2 layers of age stratification. If you have more layers, it will be increased according to the layers. Please refer to the above link and justify the required sample (or) only mention another appropriate sample size calculation techniques.

We appreciate the reviewer's clarification. The table below is our sampling table in the protocol. We have three stratifications (age groups, gender, and residential area urbanity). Since we wanted to estimate the percentage of public knowledge/practice/awareness in each stratum at the national level, each stratum should have at least 500 samples (total 2,000) according to the calculation recommended by the guide of the World Bank as below. After considering the attribution rate, it became 2,216 in total.

Age group Male Female Total

 urban rural urban rural 

18–29 125 125 125 125 500

30–39 125 125 125 125 500

40–49 125 125 125 125 500

≥50 125 125 125 125 500

Total 500 500 500 500 2,000

(Questionnaires)

Line 101 to 104: The authors stated the pretest was done. If any item analysis was done, please stated the results which will increase the validity of the questionnaires. Moreover, the study was claimed as nationwide study and to get baseline information, the authors should express detail procedure of development of the questionnaires and testing the internal and external consistency.

We appreciate the reviewer's question on this point. The pretest in the manuscript was done for the confirmation of the correctness of translation in local language. As the questionnaire is already used globally in several countries for comparing the results, we did not develop the questionnaire and did not run any statistics to test its validity and reliability 

Usually, the questionnaires contain general information of the study population as part A. I believe the authors already had that part. If so, please mention it and the data must show in the result section also. Because the readers want to know the general information of the study population.

We appreciate the reviewer's question on this point. We removed the section on general characteristics from the questionnaire in the manuscript to highlight the main point of the questionnaire. However, as we mentioned in the Method, we collected the general characteristics such as age, birthday, gender, residential areas of respondents and already provided the general information in the Result.

According to the question A1, the population will divide 2, Yes and no. Further questions are based on "Yes" sub population. It is not clear whether the authors apply questions B, C and D to all participants or only to those who answer to Yes in question A1. Please mention it clearly in the method section and data analysis section for any sub group analysis.

We appreciate the reviewer's clarification on this point. As the reviewer pointed out, we applied the questions B, C and D to only to those who answered to YES in question A1. We have added this explanation in the Method section.

Data Analysis: Thank to the authors for considering the sampling weights to reduce the bias. But need to mention any subgroup analysis and how to handle the nonresponse rate because non response rate is important for MPPS.

We appreciate the reviewer's clarification on this point. We have added the information on non-response rate in the Results (64.0%). As the panel consists of mobile phone numbers for contacting survey respondents, and Myanmar mobile connections are almost exclusively pre-paid subscriptions which are easily (and cheaply) replaceable, it is quite common to encounter numbers not in service. For mobile phone connections, not-reachable status (either because phone subscription is disconnected or phone is turned off), followed by respondent not wanting to talk (as they might be outside home attending some personal business), quality of voice making interviews difficult, call drop due to network issues, etc. are all common reasons for non-response. However, as an advantage of mobile phone panel survey, those no-responses were replaced with a different respondent with similar demographics (since demographic quota controls are in place) to satisfy the expected sample size when we faced those cases. We have added this point in the Methodology- Data collection.

Since the authors expressed the difference between groups, I suggest to calculate multivariable analysis for confounder-adjusted estimates to minimize the bias and show the strength of association.

We appreciate the reviewer's suggestion on this point. We are writing another paper to analyze the strength of association, using multivariable analysis as planned. We would like to keep this manuscript as a description of the cross-sectional survey.

Result

Again, I recommend to stick the reporting guideline. Please give the characteristics of study participants like Age, gender, residence, education, occupation, socioeconomic status etc

We appreciate the reviewer’s comment on this point. We have added the information on basic characteristics of the study partisans in the Result and table. However, to shorten the time of interview, we did not collect the information on education, occupation, and socioeconomic status. We will consider collecting the information in the next survey for more detailed analysis.

Please indicate number of participants with missing data or non-response rate. Since this is a survey type study, the readers want to know the non-response rate.

We appreciate the reviewer's clarification on this point. We have added the information on non-response rate in the Results (64.0%). As the panel consists of mobile phone numbers for contacting survey respondents, and Myanmar mobile connections are almost exclusively pre-paid subscriptions which are easily (and cheaply) replaceable, it is quite common to encounter numbers not in service. For mobile phone connections, not-reachable status (either because phone subscription is disconnected or phone is turned off), followed by respondent not wanting to talk (as they might be outside home attending some personal business), quality of voice making interviews difficult, call drop due to network issues, etc. are all common reasons for non-response. However, as an advantage of mobile phone panel survey, those no-responses were replaced with a different respondent with similar demographics (since demographic quota controls are in place) to satisfy the expected sample size when we faced those cases. We have added this point in the Methodology- Data collection.

I know the table 3,4 and 5 are representing the sub groups. I am not sure the table 6. Please specify the total number of each group in the heading of the tables and in the test also. It is confused the readers. It is better if the authors show the flow diagram of the number of the response of the participants.

We appreciate the reviewer’s comment and clarification. All table 3, 4, 5 and 6 are representing the people who know “antibiotics” (Yes in question A1, N=1,830). We have added the total number in the heading of the table and in the test.

Discussion

The whole discussion is well written and can explain the result well. Recommendation is also appropriate. But it can change after the authors adjust the data analysis. As I mention before, more Myanmar contexts/studies should be discussed here.

We appreciate the reviewer's suggestion. We have revised some parts of the Discussion accordingly. However, the challenges we have are that there are very limited number of papers/reports from Myanmar, when we search them in the Pubmed and Google Scholoar, as pieces of evidence to back up our insights. Instead, we put the comparison of the results between Myanmar and other countries in the Discussion. We hope the reviewer could understand it.

(Strength and limitation of the study)

Line 325-338, the authors want to express why they use MPPS. So, the whole para graph should express in the introduction or beginning of the method.

We appreciate and agree on the reviewer's suggestion on this point. We have revised as suggested.

The authors should address other important factors such as social and economic factors which influence the results. Any potential bias should be expressed here.

We appreciate and agree on the reviewer's suggestion. We have revised as suggested. Although we did not collect the information on those important factors, we would like to collect them in the next survey as those are factors that could influence the results and contribute to potential bias.

Reviewer #6:

The study covers important public health topic and it is very timely. The manuscript reads very well.

Some comments for the better understanding on the method applied and findings.

Method

Line number - 84-85/ 98 - Please consider giving some clarification on the phrase "representative, high-frequency panel data". And some detail on how the sample frame was identified? How the samples were randomly selected?

We appreciate the reviewer's clarification on this point. We have changed the phrase “representative, high-frequency panel data" into “representative data” to clarify the phrase. We have also added the information on the sample frame and sampling methodology in the Method. The mobile phone panel is composed of 200,000 contacts that are updated regularly. Field workers across the country interview potential respondents in person at their homes, collect their basic demographic information and receive their permission to be included in the panel to respond to several MPPSs on various topics. In this survey, the total planned sample of 2,000 was distributed equally across 16 demographic segments, which are gender (male and female), age group (18-29, 30-39, 40-49, and ≧50 years old), and residential area urbanity (urban and rural), and 125 samples were allocated in each segment for computer-assisted random sampling.

Was response rate taken into account when sample size was calculated?

We appreciate the reviewer's clarification on this point. We did not consider the non-response rate for the sample size calculation as the mobile phone panel survey can satisfy the sample size by replacing non-responders with different responders with similar demographics. We have added this point in the Methodology- Data collection.

Line number - 91 - there was a plan to do series of survey and sample size was calculated to compensate 5% attrition rate. It is suggested to elaborate more about the plan in discussion or where relevant.

We appreciate the reviewer’s comment on this point. As we have already mentioned in the Discussion, the surveys planned to be implemented in a series manner to measure the impact of the interventions such as public education and awareness campaign, that is written in the Myanmar National Action Plan for Containment Antimicrobial Resistance (2017-2022). We added this point in the Discussion.

Line 102 - Any references used in the development of the questionnaire? Did the authors use standard questionnaire and any reference on its reliability?

We appreciate the reviewer’s comment on this point. As we mentioned in the Method-Data collection, this questionnaire is the standard questioner and has been used globally like WHO and European Union Commission to compare the results among the countries. We have also put the reference number there.

Line 111 - Data were double-entered into Excel. Please include the reason of doing double entries and was there any inconsistency when/if the authors did data validation? If so, how were the errors managed?

We appreciate the reviewer's comment on this very important point. Data validation is done daily by starting from the second day of the work till the next two days of the work ending. Quality control team selects randomly 30% of achievements per interviewer and investigates either by listening to the recordings or calling the respondent for verification. We have added this point in the Methodology- Data collection.

Line 117 - Method used for calculating 95% CI was presented but no 95% CI was reported in the result session. It is suggested to include 95% CI.

We appreciate the reviewer’s comment and pointing out our error. We have added 95% CI on the estimated percentage (%) of each response both in the sentences and tables.

Important socio-economic factors such as education, income, occupation should be considered including in future surveys.

We appreciate the reviewer’s important suggestion. We will consider collecting those socio-economic factors in the future surveys to do more detail analysis.

Results

What is the response rate?

We appreciate the reviewer's clarification on this point. We have added the information on non-response rate in the Results (64.0%). As the panel consists of mobile phone numbers for contacting survey respondents, and Myanmar mobile connections are almost exclusively pre-paid subscriptions which are easily (and cheaply) replaceable, it is quite common to encounter numbers not in service. For mobile phone connections, not-reachable status (either because phone subscription is disconnected or phone is turned off), followed by respondent not wanting to talk (as they might be outside home attending some personal business), quality of voice making interviews difficult, call drop due to network issues, etc. are all common reasons for non-response. However, as an advantage of mobile phone panel survey, those no-responses were replaced with a different respondent with similar demographics (since demographic quota controls are in place) to satisfy the expected sample size when we faced those cases. We have added this point in the Methodology- Data collection.

Line 133-134 - 68.5% lived in urban area. It is different from the data presented in table 2 where 68.5% lived in rural.

We appreciate the reviewer’s pointing out our error in the manuscript. We corrected that 68.5% lived in rural area (not urban area) in the Result (Table 2 is correct).

When giving the p-values for difference groups comparison in result narrative, it is better to move the p value to the place near to variable name rather them writing all p values in the parenthesis at the end of the statement which sometimes is difficult to interpret.

We appreciate the reviewer’s suggestion. We have revised this point accordingly.

Conclusion

Line 349-351 - Prediction on the use of antibiotics during the COVID19 pandemic is not based on the current study findings and better to rephrase as a recommendation and/ or move it to discussion by giving the appropriate references.

We appreciate and agree on the reviewer’s suggestion. We have rephrased the sentence with the appropriate references and move it from Conclusion to the Discussion.

Reviewer #7

Congratulations for your well prepared manuscript.

May I suggest to add some modifications by the following comments:

Line 22-23: Paraphrase of the aim will be better if it is consistent with the title of manuscript.

We appreciate the reviewer’s suggestion. We have revised it accordingly.

Line 93: Reference for sample size consideration and formula for attrition rate should be cited.

We appreciate the reviewer’s suggestion. We have revised it accordingly.

Line 99: How are the panels framed for random selection of each participant?

We appreciate the reviewer's clarification on this point. We have also added the information on the sample frame and sampling methodology in the Method. The mobile phone panel is composed of 200,000 contacts that are updated regularly. Field workers across the country interview potential respondents in person at their homes, collect their basic demographic information and receive their permission to be included in the panel to respond to several MPPSs on various topics. In this survey, the total planned sample of 2,126 was distributed equally across 16 demographic segments, which are gender (male and female), age group (18-29, 30-39, 40-49, and ≧50 years old), and residential area urbanity (urban and rural), and 133 samples were allocated in each segment for computer-assisted random sampling.

Line 316: Awareness raising on antibiotic use, antibiotic resistance and banning of antibiotic use in farm animals should be highlighted for rural people because they occupied for about two third of Myanmar population and are exposed more to farm animals than urban residents.

We appreciate the reviewer’s comment on this very point. We also feel the same, but the results did not show any statistical differences between urban and rural residents, and the urban people also may sell the antibiotics to the rural residents. So, we concluded that awareness raising on antibiotic use, antibiotic resistance and banning of antibiotic use in farm animals is important for both urban and rural residents (not only for rural people).

Line 306-307, 332-333: Along with increasing ownership of mobile phones in Myanmar, mobile health education by mean of text messages about antimicrobials use and antimicrobial resistance should be recommended as communication channel of awareness raising campaign.

We appreciate the reviewer’s comment on this very point. However, this survey did not demonstrate that the mobile phone played an important role as a mean of health education while the TV and Facebook did. Since the respondents in the survey did not answer that the mobile phone is important as a media, we did not mention the mobile phone in the Discussion.

Reviewer #8

Summary of the research and overall impression 

This manuscript presented to access the knowledge, awareness and practice of using antibiotics among general populations in Myanmar using MPPS. Methodology session is required to revise/ add more detail information in terms of sampling, study period, data collection approach and data management. Data analysis is stated clearly. The results session is clear but the discussion and conclusions are required to improved. This study will benefit to the community and national AMR program to strategically implement the national AMR action plan. I would agree to publish this article for wider data dissemination purpose to the academic audience as well as useful for the uptake by the national program. 

It is recommended to improve the English language for academic writing and publication. For example, the use of tense (verb) in the results and discussion sessions should be past tense. The terms should be consistent: “participants” or “respondents” or “citizens” throughout the paper. Similarly, the term “overuse” or “misuse” should be consistent. If “overuse” and “misuse” are different definition, please elaborate in methods session. 

We appreciate the reviewer’s comment. We revised the manuscript on those points, although we received professional English proofreading from two native academic editors. In terms of “overuse” and “misuse”, we used those terms based on the dictionary definition (overuse: excessive use/ misuse: wrong use). Since this definition is based on the language dictionary, we did not elaborate the diminution in the manuscript.

Discussion on specific area of improvement 

Abstract

1. The information presented in the abstract is clear, concise and complete. Thank you. Please add how the participants were randomly selected from which sampling frame. It said “from the panel” – please elaborate more what is the “panel” represent for. 

We appreciate the reviewer's clarification on this point. Since the abstract has a limitation of the word number, we have also added the information on the sample frame and sampling methodology in the Method. The mobile phone panel is composed of 200,000 contacts that are updated regularly. Field workers across the country interview potential respondents in person at their homes, collect their basic demographic information and receive their permission to be included in the panel to respond to several MPPSs on various topics. In this survey, the total planned sample of 2,216 was distributed equally across 16 demographic segments, which are gender (male and female), age group (18-29, 30-39, 40-49, and ≧50 years old), and residential area urbanity (urban and rural), and 133 samples were allocated in each segment for computer-assisted random sampling.

Introduction 

2. Add the key findings from the 3 countries in SEA region (India, Indonesia and Thailand) – eg. the proportion of public awareness and understanding about AMR in these counties. 

We appreciate the reviewer's suggestion on this point. We have added brief highlights from ARM awareness surveys of WHO South-East-Asian region. Detail findings are also explained in the “Discussion” as a comparison of findings in between those countries and Myanmar.

3. List the countries where the WHO AMR nationwide survey was conducted in the SEA region. 

We appreciate the reviewer's comment on this point. As we mentioned in the Introduction, in SEA region, India and Indonesia implemented the WHO survey by online, whereas Thailand conducted a survey via the creation of a module to measure awareness as part of the 2017 national health welfare survey.

4. Mention additional information about the national AMR program in Myanmar and the action/ activities implementing for AMR strategic action plan. 

We appreciate the reviewer's comment on this point. We have added a prevalence of AMR in Myanmar (WHO priority pathogens) from the national AMR report published in 2019, highlighting the magnitude of AMR and AMR activities in Myanmar that were planned in the AMR action plan.

Methods

5. Explain the sampling procedure – how the participants were chosen from the panel data nationwide. 

We appreciate the reviewer's clarification on this point. We have also added the information on the sample frame and sampling methodology in the Method. The mobile phone panel is composed of 200,000 contacts that are updated regularly. Field workers across the country interview potential respondents in person at their homes, collect their basic demographic information and receive their permission to be included in the panel to respond to several MPPSs on various topics. In this survey, the total planned sample of 2,216 was distributed equally across 16 demographic segments, which are gender (male and female), age group (18-29, 30-39, 40-49, and ≧50 years old), and residential area urbanity (urban and rural), and 133 samples were allocated in each segment for computer-assisted random sampling.

6. Table 1 – please move it to Annex. 

We appreciate the reviewer’s suggestion. We moved it to the Annex.

7. Clarify the “period of the study” and “study areas” included in the study where the participants were enrolled in the survey.

We appreciate the reviewer’s suggestion. We have added the information in the Results.

8. Suggest to mention about “data management”, such as data cleaning and checking, how the data was stored for such a certain period of time, data security measure, etc.

We appreciate the reviewer’s comments. We did not put the information on data management in the manuscript that is too much for the journal paper, but the information on data management is written in the research protocol approved by the Ethics Review Board. In this study, data entry and modification were performed by a pre-determined person in charge using Microsoft EXCEL. Records of revisions, if any, was recorded at the National Health Laboratory, the Ministry of Health, Myanmar, and will be retained until the retention period (five years). Information on media is strictly managed in a lockable cabinet at the National Health Laboratory, the Ministry of Health, Myanmar, where access is controlled. The electronic media itself is managed with a user ID and password. The electronic media itself is managed with user IDs and passwords. Only principal investigators and collaborators approved by the Ethics Review Board have access to these research data. 

9. Add the information about non-response rate, which is important for mobile phone interviews using IVR at the call center. 

We appreciate the reviewer's clarification on this point. We have added the information on non-response rate in the Results (64.0%). As the panel consists of mobile phone numbers for contacting survey respondents, and Myanmar mobile connections are almost exclusively pre-paid subscriptions which are easily (and cheaply) replaceable, it is quite common to encounter numbers not in service. For mobile phone connections, not-reachable status (either because phone subscription is disconnected or phone is turned off), followed by respondent not wanting to talk (as they might be outside home attending some personal business), quality of voice making interviews difficult, call drop due to network issues, etc. are all common reasons for non-response. However, as an advantage of mobile phone panel survey, those no-responses were replaced with a different respondent with similar demographics (since demographic quota controls are in place) to satisfy the expected sample size when we faced those cases. We have added this point in the Methodology- Data collection.

10. Mention how the data quality check was conducted (in a real-time?) for IVR and mobile phone interviews. Any follow up questions if the data is incomplete or occurs with errors. 

We appreciate the reviewer's comment on this very important point. Data validation is done daily by starting from the second day of the interviews till the next two days of the interviews ending. Quality control team selects randomly 30% of achievements per interviewer and investigates either by listening to the recordings or calling the respondent for verification. We have added this point in the Methodology- Data collection.

Ethical considerations 

11. Line 126 – please double check the name of the institutional board for ethical review. I assume it should be “Institutional Review Board”, not “Institutional Research Board”. 

We appreciate the reviewer's pointing out our error. We corrected the name of the board.

Results 

12. Move the study period to the methods session. 

We appreciate the reviewer’s suggestion. We move the study period to the Method session.

13. The sample size (in the methods session) said 1940 participants, but how it was ended up with 2045 participants. Please explain in the methods session and in results session, total sample size is enough to present. 

We appreciate the reviewer’s comment. We have added the explanation in the Method and Results session. The required sample size was 2,216 based on the calculation. However, less participant cooperated the interview through multiple interviewers, and finally, total participants became 2,045, that was less than required sample size. This point is also mentioned as a study limitation in the Discussion.

14. Line 133 – 68% lived in RURAL areas (according to the data from the table). Please correct. 

We appreciate the reviewer’s pointing out our error in the manuscript. We corrected that 68.5% lived in rural area (not urban area) in the Result (Table 2 is correct).

15. Mention the proportion of participants in each state/ region in the Annex table. 

We appreciate the reviewer’s comment. We have added the information on the proportion of participants in each state and region in the Result (table).

Discussion and conclusions

16. Line 226 – “excellent representation” is not the academic writing; reflecting cannot measure how much “excellent” stands for. Please mention scientifically for the representativeness of the sample populations from the study. 

We appreciate and agree on the reviewer’s comment. We have rephrased the sentence.

17. In general, add the data from other literatures/ studies when explaining about the comparisons of data from this study findings, rather than mentioning “less likely” or “more likely”, so that the readers can get more detail understanding of how the differences is small or large in this study, compared to other studies. 

We appreciate the reviewer’s comment. We have added the detailed differences as compared to the results in other countries’ survey (references), where we used “less likely” and “more likely”.

18. Line 242-245 (…nationwide dissemination……multiple communication channels.) – how did you get this recommendation to be accomplished the knowledge dissemination in Myanmar. Please refer to and double check with the findings that can reference for this recommendation. 

We appreciate the reviewer’s comment. This recommendation was derived from this survey results; i) the differences in the knowledge and awareness of antibiotics/AMR among the age groups, gender, and residential urbanity, and ii) the multiple sources of antibiotics/AMR information like medical professionals, friends, TV and Facebook.

19. Line 266-267 (…in reality……without prescription.) – this statement is not included in the findings. Please add the reference if relevant or please remove the statement if that is not from the findings or no reference can be added. You may not want to add information without any evidence or reference in the discussion. 

We appreciate the reviewer’s comment. We have added the references for the statement from other studies in Myanmar.

20. Line 309 – you cannot state “people from Myanmar” because this study was conducted in a sample population of Myanmar. You can state that “in this study, participants were less likely….”

We appreciate the reviewer’s suggestion. We have revised it accordingly.

21. Line 317-319 – suggest to mention that further collaboration with the agriculture, livestock and veterinary department for providing proper training and education program to the farmers and livestock, breeding farms regarding the AMR. 

We appreciate the reviewer’s practical suggestion. We have revised it accordingly.

22. Limitation: mention the weakness of mobile phone survey, instead to face-to-face survey (eg. non-response rate). 

We appreciate the reviewer’s comment on the limitation. We have revised it, accordingly, including non-response rate.

23. Please mention anything about the safeguarding measures applied throughout the MPPS survey approach. If not, add this as the limitation. 

We appreciate the reviewer’s comments on the safeguarding measures of the data in this MPPS survey. 

All the interviewers were trained based on the study protocol. Our study protocol highlighted the duty to report, even including anonymously, and the confidential channel to do so. It also highlighted the fact that the interviewer should not ask any questions related to the matter but ask permission to share the personal contact information of the interviewee. As a data protection, information on media is strictly managed in a lockable cabinet at the National Health Laboratory, the Ministry of Health, Myanmar, where access is controlled. The electronic media itself is managed with a user ID and password. The electronic media itself is managed with user IDs and passwords. Only principal investigators and collaborators approved by the Ethics Review Board have access to these research data. 

24. However, the later part of the paragraph (line 327 onwards) stated about the advantage of MPPS. In this case, please present clearly “limitation” and “strength” of the survey. In addition, please also add the literature and reference about using MPPS methodology. Reference from Oxfam related to CATI (using phone interview) is provided for additional reference as necessary. 

We appreciate the reviewer’s comments. We have made the paragraph concise to present the limitation and strength clearly with adding the references and move some parts up to the Introduction for better manuscript structure.

Conclusion 

25. COVID-19 related AMR situation is just appeared in the conclusion. It is agreeable to include and discuss about this survey conducted before COVID-19 pandemic. Thus, include or move some information about COVID-19 related AMR situation in the discussion first, then present it again to conclude how this data could be impacted after the COVID-19 pandemic. 

We appreciate the reviewer’s suggestion. We have rephrased the sentence with the appropriate references and move it from Conclusion to the Discussion.

26. The facts in conclusion should be more concisely presented one by one, rather than general presentation. 

We appreciate the reviewer’s suggestion. We have revised the Conclusion accordingly, reflecting other reviewers’ comments.

---

## [Decision Letter · Decision Letter 1]

8 Aug 2022

Public knowledge, practices, and awareness of antibiotics and antibiotic resistance in Myanmar: The first national mobile phone panel survey

PONE-D-22-06745R1

Dear Dr. Miyano,

We’re pleased to inform you that your manuscript has been judged scientifically suitable for publication and will be formally accepted for publication once it meets all outstanding technical requirements.

Kind regards,

Monica Cartelle Gestal, PhD

Academic Editor

PLOS ONE

Additional Editor Comments (optional):

Reviewers' comments:

Reviewer's Responses to Questions

**Comments to the Author**

1. If the authors have adequately addressed your comments raised in a previous round of review and you feel that this manuscript is now acceptable for publication, you may indicate that here to bypass the “Comments to the Author” section, enter your conflict of interest statement in the “Confidential to Editor” section, and submit your "Accept" recommendation.

Reviewer #2: All comments have been addressed

Reviewer #4: All comments have been addressed

Reviewer #5: All comments have been addressed

Reviewer #7: All comments have been addressed

Reviewer #8: All comments have been addressed

2. Is the manuscript technically sound, and do the data support the conclusions?

Reviewer #2: Yes

Reviewer #4: Yes

Reviewer #5: Partly

Reviewer #7: Yes

Reviewer #8: Yes

3. Has the statistical analysis been performed appropriately and rigorously? 

Reviewer #2: Yes

Reviewer #4: Yes

Reviewer #5: N/A

Reviewer #7: Yes

Reviewer #8: Yes

4. Have the authors made all data underlying the findings in their manuscript fully available?

Reviewer #2: Yes

Reviewer #4: Yes

Reviewer #5: No

Reviewer #7: Yes

Reviewer #8: No

5. Is the manuscript presented in an intelligible fashion and written in standard English?

Reviewer #2: Yes

Reviewer #4: Yes

Reviewer #5: Yes

Reviewer #7: Yes

Reviewer #8: Yes

6. Review Comments to the Author

Reviewer #2: As the author have corrected according to the comments, my recommendation for this paper is "Accept".

Reviewer #4: Reviewer’s comment for R1`

Overall comment:

The authors thoroughly revised the R1 and answers almost satisfactorily to the reviewer’s comments and suggestions. On my point of view, only the minor correction is needed for further revision.

Major Comments

Title and Methodology:

- the authors explained well how to achieve representative samples for nationwide distribution in their point by point response. The authors changed the “ nationwide” to “national” in their R1 manuscript

- I’d like to suggest the wording “nationwide MPPS” is more appropriate than “national” but the authors should mention the selected 12 States and Regions out of total 15 in Myanmar in the “Methodology” section. And should mention reasons of excluding Kayah, Chin and Rakhine in “Methodology” or “Limitations of the study” section.

- The authors explained well about the above mentioned points in the point by point response to my comments and also mentioned the selected States and Regions in the table/info sheet. However, these facts should be clearly mentioned in the text of manuscript, so that the readers can clearly see these facts and it gives the strength for “nationwide survey” research article.

- The authors explained clearly the sampling frame in the response as “ planned sample 2216 was distributed 16 demographic segments which are gender, age and urbanity. Thus 133 samples were collected from each segment ……………

- These facts should be included in the methodology section to be able to make clear description to the readers.

Minor comments

Abstracts

- Line 20-22 should be divided into two sentences instead of using long sentence. “In 2017… Ministry of Health and Sports, Myanmar. One of its objectives…”

Ethical Considerations

- Line 162, correct as “Institutional Review Board for Biomedical Research involving human subjects”

Reviewer #5: Although the authors have responded almost all questions adequately, the nature of the study is descriptive study. I have no more comments.

Reviewer #7: Thank you for your thorough reply to each of my comments. Your well updated manuscript is hope to be published soon.

Reviewer #8: Thank you the first author and co-authors addressing all the comments from the many reviewers. Really appreciate to produce this paper which is important for Myanmar AMR response. I accept the revision and no further comment.

7. PLOS authors have the option to publish the peer review history of their article (what does this mean?). If published, this will include your full peer review and any attached files.

Reviewer #2: No

Reviewer #4: No

Reviewer #5: No

Reviewer #7: No

Reviewer #8: **Yes: **Poe Poe Aung

---

## [Editor Report · Acceptance letter]

10 Aug 2022

PONE-D-22-06745R1 

Public knowledge, practices, and awareness of antibiotics and antibiotic resistance in Myanmar: The first national mobile phone panel survey 

Dear Dr. Miyano:

I'm pleased to inform you that your manuscript has been deemed suitable for publication in PLOS ONE. Congratulations! Your manuscript is now with our production department. 

Kind regards, 

on behalf of

Dr. Monica Cartelle Gestal 

Academic Editor

PLOS ONE